# Total Oxidation of Methane on Oxide and Mixed Oxide Ceria-Containing Catalysts

Marius Stoian [1], Vincent Rogé [2], Liliana Lazar [3], Thomas Maurer [4], Jacques C. Védrine [5], Ioan-Cezar Marcu [1,6] and Ioana Fechete [7,8,*]

1   Laboratory of Chemical Technology and Catalysis, Department of Organic Chemistry, Biochemistry and Catalysis, Faculty of Chemistry, University of Bucharest, Blv. Regina Elisabeta 4-12, 030018 Bucharest, Romania; marius_stoian7@yahoo.ro (M.S.); ioancezar.marcu@chimie.unibuc.ro (I.-C.M.)
2   Luxemburg Institute of Science and Technology, 4362 Esch-Sur-Alzette, Luxembourg; vincent.roge@list.lu
3   Department of Chemical Engineering, "Cristofor Simionescu" Faculty of Chemical Engineering and Environmental Protection, "Gheorghe Asachi" Technical University of Iasi, Prof. dr. docent Dimitrie Mangeron Street 73, 700050 Iasi, Romania; lillazar@ch.tuiasi.ro
4   Laboratoire Lumière, nanomatériaux et nanotechnologies–L2n, Université de Technologie de Troyes & CNRS ERL 7004, Rue Marie Curie 12, 10000 Troyes, France; thomas.maurer@utt.fr
5   Laboratoire de Réactivité de Surface, Université Pierre et Marie Curie, Sorbonne Université, UMR-CNRS 7197, 4 Place Jussieu, 75252 Paris, France; jacques.vedrine@sorbonne-universite.fr
6   Research Center for Catalysts and Catalytic Processes, Faculty of Chemistry, University of Bucharest, Blv. Regina Elisabeta 4-12, 030018 Bucharest, Romania
7   ICD-LASMIS, Université de Technologie de Troyes, Antenne de Nogent, Pôle Technologique de Sud Champagne, 52800 Nogent, France
8   Nogent International Center for CVD Innovation-NICCI, LRC-CEA-ICD-LASMIS, Université de Troyes-Antenne de Nogent, Pôle Technologique Sud Champagne, Rue Lavoisier 26, 52800 Nogent, France
*   Correspondence: ioana.fechete@utt.fr

**Abstract:** Methane, discovered in 1766 by Alessandro Volta, is an attractive energy source because of its high heat of combustion per mole of carbon dioxide. However, methane is the most abundant hydrocarbon in the atmosphere and is an important greenhouse gas, with a 21-fold greater relative radiative effectiveness than $CO_2$ on a per-molecule basis. To avoid or limit the formation of pollutants that are dangerous for both human health and the atmospheric environment, the catalytic combustion of methane appears to be one of the most promising alternatives to thermal combustion. Total oxidation of methane, which is environmentally friendly at much lower temperatures, is believed to be an efficient and economically feasible way to eliminate pollutants. This work presents a literature review, a statu quo, on catalytic methane oxidation on transition metal oxide-modified ceria catalysts ($MO_x/CeO_2$). Methane was used for this study since it is of great interest as a model compound for understanding the mechanisms of oxidation and catalytic combustion on metal oxides. The objective was to evaluate the conceptual ideas of oxygen vacancy formation through doping to increase the catalytic activity for methane oxidation over $CeO_2$. Oxygen vacancies were created through the formation of solid solutions, and their catalytic activities were compared to the catalytic activity of an undoped $CeO_2$ sample. The reaction conditions, the type of catalysts, the morphology and crystallographic facets exposing the role of oxygen vacancies, the deactivation mechanism, the stability of the catalysts, the reaction mechanism and kinetic characteristics are summarized.

**Keywords:** methane; total oxidation; cerium; oxide catalysts; environment

## 1. Introduction

Volatile organic compounds (VOCs) are environmental pollutants regarded as precursors for the formation of tropospheric ozone (a greenhouse gas), the depletion of stratospheric ozone and the formation of photochemical smog in urban environments. The emission of VOCs by human activities and various industrial processes is a serious

source of air pollution and is therefore a problem for human health and the environment in general [1–10]. Among VOCs, methane ($CH_4$), as a potent greenhouse gas, is the most abundant reactive trace gas in the atmosphere. Greenhouse gases such $CO_2$ and $CH_4$ are responsible for global warming and have become a serious environmental problem [11–13].

Methane, one of the most abundant greenhouse gases after water vapor and carbon dioxide, whose atmospheric concentration is close to that of krypton, is a greenhouse gas that has seen the fastest increase in concentration in the industrial era. Its concentration rose since the beginning of the 19th century from 0.7 to 17 ppmv (parts per million by volume), while that of carbon dioxide increased from 280 to 380 ppmv. This increase is mainly due to the growing exploitation of methane sources as human activities demand more energy, which currently represent approximately two-thirds of the 500 million tons emitted each year and increasingly by ~1% annually [14–17].

$CH_4$ arises from both natural and anthropogenic sources, but anthropogenic emissions have become more important than natural emissions. Among the sources linked to human activity, we can highlight the combustion of fossil fuels and biomass, paddy fields, animal husbandry, industrial leaks and garbage dumps. Following the industrial revolution, $CH_4$ was the source of a radiative disturbance equivalent to one-third of that of carbon dioxide. However, there are also indirect effects: methane is indeed oxidized in the atmosphere, which leads to an increase in the water vapor content in the "upper" atmosphere and therefore an increase in the greenhouse effect. In the "lower" atmosphere, reactions destroying methane lead to an increase in the ozone content. If these indirect effects are included, methane would be responsible for radiative forcing that is half that of carbon dioxide. Although $CH_4$ lifetime in the atmosphere is much shorter than that of $CO_2$, $CH_4$ is more efficient at trapping radiation than $CO_2$. Note that the tetrahedral conformation of the methane molecule leaves it more latitude to vibrate than the linear carbon dioxide molecule, whose atmospheric concentration is two orders of magnitude higher. Thus, for equal amounts, methane absorbs much more infrared radiation than carbon dioxide. However, the warming potential of methane is 21-times higher than that of carbon dioxide at equivalent emission rates [16–18].

Considering that $CH_4$ is a powerful greenhouse gas, it is a challenge to reduce or limit the amount of $CH_4$ emitted into the atmosphere due to its effect on global warming according to current and future worldwide regulations. Several methods exist to limit $CH_4$ emissions, such as adsorption, membrane separation and combustion [19,20].

The combustion process was defined by Alessandro Volta (1745–1827), who also discovered methane in 1776. Volta even designed a gun running on methane. The combustion of methane can be performed at high temperatures, which is appealing for conventional thermal combustion, and at low temperatures in the presence of solid catalysts, which is appealing for catalytic combustion or oxidation. The conventional thermal combustion of methane presents important drawbacks, such as very high temperatures (up to 1500 °C) for total oxidation also resulting in the production of NOx as by-product. The $CH_4$ molecule consists of strong C–H bonds that are difficult to dissociate, requiring high temperatures in conventional thermal combustion [21]. The catalytic combustion of $CH_4$ (and hydrocarbons) has received considerable attention due to its low temperature and practical applications in both power generation and pollutant abatement. The reaction may be represented by the equation:

$$CH_4 + 2\,O_2 \rightarrow CO_2 + 2\,H_2O \tag{1}$$

Other reactions may also be involved to a greater or lesser extent. These reactions could include steam reforming and water gas shift reactions:

$$CH_4 + 3/2\,O_2 \rightarrow CO + 2\,H_2O \tag{2}$$

$$CH_4 + H_2O \rightarrow CO + 3\,H_2 \tag{3}$$

$$2\,H_2 + O_2 \rightarrow 2\,H_2O \tag{4}$$

$$CO + H_2O \rightarrow CO_2 + H_2 \tag{5}$$

$CH_4$ has a relatively strong C–H bond (450 kJ/mol) [21] and is the most difficult alkane to oxidize completely. Therefore, the catalytic oxidation of methane is the best approach for the dissociation of C–H bonds, the oxidation temperature being thus considerably reduced [22,23]. It has been confirmed that the catalytic oxidation of methane is an efficient method for capturing low-concentration methane [24,25]. This environmentally friendly technology offers the possibility to produce heat and energy at much lower temperatures than conventional thermal combustion. Moreover, catalytic combustion or oxidation is viewed as a viable approach to resolve this important environmental problem, at least in the short term [21,23,26].

As a means of reducing $CH_4$ emissions, a major challenge for the environment, the total catalytic oxidation of $CH_4$ has been extensively studied [27–31]. The complete oxidation of $CH_4$ into $CO_2$ (a less potent greenhouse gas), particularly at low temperatures (below 500 °C), has been widely studied [32–34]. Since $CH_4$ has the lowest carbon to hydrogen ratio among all hydrocarbon fuels, it generates the lowest amount of $CO_2$ per unit of produced energy during combustion.

For complete oxidation applications, very active and highly stable catalysts are needed. Extensive efforts were made earlier towards the low-temperature combustion of $CH_4$ using different noble metals, metal oxides and mixed metal oxides acting as catalysts [23,32,33,35–37].

Noble metal catalysts [28,38–44], particularly Pd, which is highly dispersed over support oxide catalysts, are known to have the highest activity for $CH_4$ oxidation at low temperature and are widely used in catalytic $CH_4$ abatement systems [28,45–53]. They showed high methane combustion activity at low temperature, above 400 °C, with deactivation occurring above 700 °C due to PdO decomposition [54,55]. Although noble metal catalysts have higher activity (per site) than metal oxide catalysts, they suffer from disadvantages, such as higher volatility and poor economics/availability. Indeed they have poor stability and tend to deactivate severely during operation [50,56–58]. Rapid deactivation under reaction conditions induced by active phase sintering [59,60] and the low thermal stability and high poisoning susceptibility of noble metal catalysts limit their effective application in the catalytic oxidation of $CH_4$.

Intensive research has been conducted to identify alternative catalysts. Among them, metal-oxide catalysts have received much attention. Although they are less active, transition metal oxides seem to be an attractive alternative to noble metal catalysts, as they are cheaper and more resistant to poisoning [25,28,39,60].

Copper oxide is one of the most effective metal oxide-based catalysts used in the total oxidation of $CH_4$, a key process involved in air pollution abatement [55,61]. However, its catalytic performance is far below that of noble metal-based systems, especially Pd and Pt, which in turn are less attractive from an economic point of view [37]. Further improvements in the catalytic performance of CuO will certainly lead to a better understanding and control of the key parameters governing its properties, particularly using supported CuO phases whose supports exhibit controlled acid-base properties. This latter feature has already been shown to be a key factor controlling the catalytic activity of supported noble-metal catalysts in the total oxidation of methane [62–64]. For example, Yoshida et al. [62] showed that palladium catalysts supported on oxides with moderate acid strength achieved higher methane conversion than palladium supported on highly acidic or basic supports. Several studies also link the acid-base properties of Cu-based catalysts to their activity in the complete oxidation of methane. Thus, Lee et al. [65] showed that CuO-supported catalysts are more active in methane combustion with acidic zeolite supports than with nonacidic silica or silicalite supports. Additionally, the catalytic activity increases when the silica-to-alumina ratio increases in zeolites, corresponding to an increase in acid strength. The authors concluded that protons could activate hydrocarbons, allowing easier oxidation. A correlation was also established between catalytic activity in both methane and propane combustion and the acidity of copper-exchanged ZSM-5, ZSM-11 and ZSM-48 catalysts [66].

M'Ramadj et al. [67] observed the adsorption of both methane and oxygen on the acid sites of Cu/ZSM-5 catalysts and concluded that methane molecules could be activated on Brønsted acid sites and that $Cu^{2+}$ species acted as Lewis acid sites. These studies show that the dependence of catalytic performance on the acid-base properties is rather complex. Nevertheless, the adsorption and activation of hydrocarbons over oxide-based catalysts, as well as the desorption of the reaction products, is related not only to the strength and distribution of the metallic cations acting as Lewis acid sites but also to those of the lattice oxygen anions acting as Lewis basic sites [68,69]. For example, the interaction of $CH_4$ with an acid-base pair on the catalyst surface results in heterolytic C–H bond breaking and the formation of $CH_3^-$ and $H^+$ species chemisorbed on the acid and base sites, respectively [69]. Stronger acid sites enhance the interaction with $CH_3^-$ carbanions and therefore favour surface-catalysed combustion [69].

Cobalt oxides are widely studied for the total oxidation of methane [61,70]. It was observed that the morphology of nanostructured cobalt influenced methane catalytic oxidation. The $Co_3O_4$ nanostructures [71] prepared by a hydrothermal approach exhibited the following characteristics: nanosheets predominantly possessed exposed (112) planes, nano-cubes possessed exposed (011) planes, and nanobelts possessed exposed (001) facets. Highly porous $Co_3O_4$ nanorods are prepared by a simple hydrothermal method and compared to bulk $Co_3O_4$ prepared by the thermal decomposition of cobalt nitrate [72]. $Co_3O_4$ nanorods showed a higher activity for $CH_4$ combustion, particularly for high GSHV ($100,000 \ h^{-1}$) tests, due to their larger surface area and advanced porous structure. Co catalysts have shown high activity but low stability [73].

Among mixed oxides, perovskites or pyrochlore-type oxides [74–79], as well as transition metal-containing mixed oxides obtained from layered double hydroxides (LDH) precursors [80–85], proved to have high activity and good stability in methane combustion. It has been observed that the promotion of lanthanum oxide and $Co_3O_4$ or $Co_3O_4$–$ZrO_2$ by cerium increases the activity of these catalysts [86,87].

As a support for noble metals (mainly Pd) [88], simple oxide [89], mixed oxides (e.g., $ZrO_2$) [90–93] or supported nanoparticles [94], ceria enhances the catalytic properties in combustion processes. Activation of oxygen on ceria gives rise to at least two species, superoxide ($O_2^-$) and peroxide ($O_2^{2-}$) ions, both known to be involved in the total oxidation of hydrocarbons [89]. Moreover, the modification of ceria with various cations is known to affect oxygen mobility, which plays an important role in the catalytic combustion of methane [93,94] and improves stability towards sintering and the oxidation activity of the resulting catalysts. This modification leads to changes in redox properties and the creation of oxygen vacancies, both of which improve the oxygen exchange capacity between the gas phase and the catalyst and its oxygen storage capacity. In this sense, considerable attention has been given to incorporating different valence cations into the ceria lattice. The oxygen atoms/vacancies attached to reducible elements are mobile and contribute to the oxygen storage/release capacity of the mixed oxides and, hence, to their oxidation activity. Other rare earth cations, such as Nd [76], Sm [76–78,95,96], Eu [78], Gd [77] and Tb [78], have been proven to increase the activity and/or thermal stability of combustion catalysts.

In recent years, $CeO_2$-based materials have been extensively studied as heterogeneous catalysts for a wide range of well-established and emerging applications [97–102]. They have been used in ceramics, glasses, fuel cells, and microelectronics and as water gas shift catalysts, three-way catalysts and hydrogenation catalysts and have found applications such as the conversion of aromatic compounds, petroleum cracking, CO and $CO_2$ conversion, and organic chemical synthesis, reflecting their importance in enhancing the performances of those systems [103–113].

Cerium oxide, $CeO_2$, is a very promising catalyst thanks to the capability of Ce to rapidly change its oxidation number from $Ce^{4+}$ to the $Ce^{3+}$ state, with a subsequent fast release of oxygen from its lattice to the nearby species [114–116]; this oxygen mobility was proven to assist Pd in surface oxygen bonding, thus helping oxidation. An interesting property of $CeO_2$ is its ability to release and absorb oxygen under alternating redox

conditions and hence to function as an oxygen buffer. As a part of car exhaust gas clean-up catalysts, $CeO_2$ widens the "lambda window" in which the catalyst can simultaneously catalyse the reduction of NO and the oxidation of CO and hydrocarbons [2,117,118]. Efforts to increase the oxygen storage capacity of $CeO_2$ by the introduction of cation dopants have been successful [119]. Several samples of cerium oxide-based solid solutions have been prepared with a variety of dopant cations, e.g., Ba, Ca, Co, Cu, Mn, Nd, Pb, Sr, Y, Zn, and Zr. Most of these materials showed enhancement in both the oxygen vacancy concentration and oxygen storage capacity as well as redox activities compared to those of undoped $CeO_2$ [119]. The purpose of doping $CeO_2$ with dopants with a valence lower than $4^+$ is to create oxygen vacancies in the $CeO_2$ structure, in contrast to the more conventional doping of $CeO_2$ with $ZrO_2$, which aims to increase its thermal stability. The properties of the oxygen vacancies, e.g., the local atomic structure, chemical environment, and binding energies, can in principle be controlled by the choice of dopants [120]. All these applications are based on the potential redox chemistry between $Ce^{3+}$ and $Ce^{4+}$, its high oxygen affinity and the absorption/excitation energy bands associated with its electronic structure. The redox property and Lewis acid and base sites on ceria make this oxide suitable for use as a catalyst or a support for chemical reactions. It must be noted that ceria is the best polishing agent for most glass compositions [121]. A significant portion of cerium products are applied annually in the polishing industry. Nanoceria with different sizes and size distributions (e.g., 30–50 nm [122] and 10–80 nm [123]) have been synthesized and investigated for shallow trench isolation chemical mechanical polishing. The best polishing performance of this nano-grade ceria was 2258 Å/min for oxide films and 220 Å/min for nitrides [124].

## 2. Ceria Generalities

Rare-earth oxides have been widely explored in catalysis, metallurgy, fuel cells, medical applications and ceramics. Cerium oxide is one of the most important rare earth oxides being actively investigated, especially its use in catalysis [97]. Cerium is a well-known rare earth element and the most abundant, although the exact average concentration of cerium in the Earth's crust is unknown. The content ranges from 10 to 300 ppm [97,125]. Rare earths are moderately abundant elements in Earth's crust that occur in a large number of minerals. Elemental cerium was first discovered from a mineral named "cerite" by Jons Jakob Berzelius and Wilhelm Hisinger in Sweden [97] and is in the form of oxides in most cases. Its elemental distribution differs in both minerals and locations. The cerium content in bastnasite is 49.1% with respect to all the rare earth contents in bastnasite from the Mountain Pass, California, U.S., while the cerium content in bastnasite from Bayan Obo, Inner Mongolia, China is 50.0%. The cerium content in monazite minerals from North Staradbroke, Australia is 45.8% and from East Coast Brazil is 47% [126].

### 2.1. Structure

Cerium belongs to the lanthanide group and has a $4f^2\,5d^0\,6s^2$ electron configuration. Cerium metal is thermodynamically unstable, in its metallic form, in the presence of oxygen [37]. Depending on the temperature and the partial pressure of oxygen, it will rapidly oxidize in a form ranging from $Ce^{3+}$(III) ($Ce_2O_3$) to $Ce^{4+}$(IV) ($CeO_2$). The most stable form of cerium oxide is $CeO_2$.

The dioxide form, $CeO_2$, has a fluorite-type structure with a face-centred cubic unit cell and space group of *Fm-3 m* [98]. In each unit cell, the lattice constant is 5.411 Å; each cerium ion is surrounded by eight equivalent oxygen anions, and each anion is tetrahedrally coordinated by four cerium cations (Figure 1). The eight-coordination sites are alternately empty and occupied by a cerium cation. The structure can also be described as a simple cubic lattice of $O^{2-}$ in which 50% of the cubic sites are occupied (Figure 2). The coordination of cerium will therefore be 8 and that of oxygen will be 4. This shows that there are large vacant octahedral holes in the structure, and this feature plays an important role in the applications of cerium oxide.

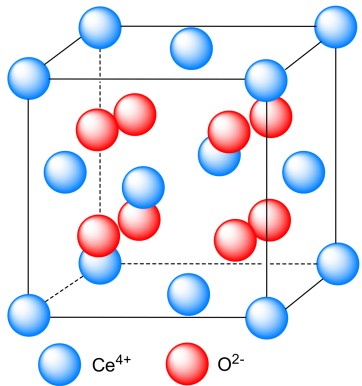

**Figure 1.** Atomic configuration of the fluorite structured $CeO_2$ unit cell. Adapted from [98].

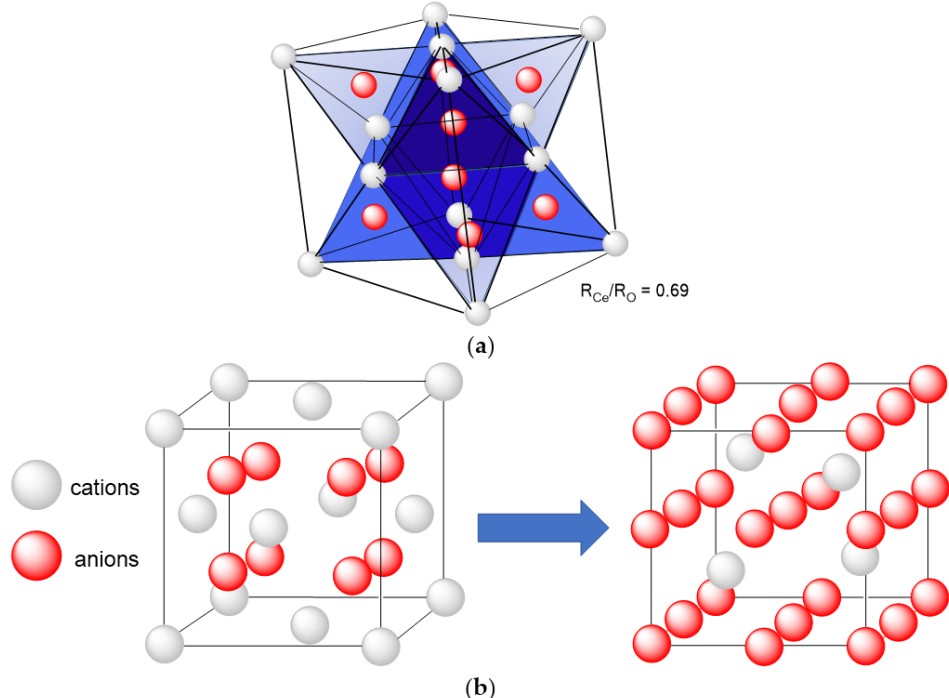

**Figure 2.** (**a**) Face-centred cubic stack of $Ce^{4+}$ (white) with 100% $T_d$ sites occupied by $O^{2-}$ (red) and (**b**) simple cubic lattice of $O^{2-}$ with 50% of the cubic (eight-coordinated) sites occupied with $Ce^4$. Adapted from [98].

Trivalent cerium (III) oxide ($Ce_2O_3$) also exists under certain conditions. $Ce_2O_3$ is unstable towards oxidation and is oxidized as pressure increases up to $10^{-40}$ atm of oxygen, where $CeO_2$ starts to form [98].

Other phases of cerium oxide have also been observed. For instance, α-phase cerium oxide, a disordered nonstoichiometric fluorite-related phase, is stable above 685 °C ($CeO_x$, $1.714 < x < 2$) [127–129], and a so-called β-phase with a rhombohedral structure ($CeO_x$, $1.805 < x < 1.812$) forms at room temperature and stays stable until 400 °C [130,131].

### 2.2. Oxygen Vacancy Defects

Intrinsic defects are present in ceria due to the alternative occupation and, thus, absence of cerium cations in the eight coordination sites. These defects can also be created by reactions with solids or the atmosphere [98]. Three different types of internal/subsurface defects observed in ceria include Frenkel (cation) defects, anti-Frenkel (anion) defects, and Schottky defects. Frenkel (or interstitial-vacancy pair) defects form when an atom or ion leaves its place in the lattice, occupies an interstitial site in a nearby location and creates a vacancy at the original site. Schottky defects form when oppositely charged ions leave their

lattice sites, creating vacancy sites, and these vacancies form according to stoichiometric units to maintain an overall neutral charge in the ionic solid (Figure 3 [132]). In ceria, the energies of cation Frenkel defects (8.86 eV/per defect) and Schottky defects (3.33 eV) are higher than those of Frenkel oxygen defects (2.81 eV); hence, the most likely form of intrinsic disorders are Frenkel-type oxygen defects [133]. It has been reported that the predominant defects in ceria (and yttria-doped ceria) are anion vacancies [134], and the concentration of interstitial Ce defects was less than ~0.1% of the total defect concentration in $CeO_{1.91}$ [135]. Anion Frenkel-type oxygen defects lead to the formation of pairs of oxygen vacancies and oxygen atoms in interstitial positions. These defects do not change the stoichiometric composition and usually have a low concentration. However, ceria can form a high concentration of vacancy defects by removing oxygen ions under a reducing environment, which induces the stoichiometry change from $CeO_2$ to $CeO_{2-x}$ (0 < x < 0.5). In this case, the created oxygen vacancies need to compensate for the negative charges formed while removing oxygen. After oxygen is removed from the $CeO_2$ lattice, the remaining electrons are associated with the change in the charge of the two cerium atoms from +4 to +3. This process is illustrated in Equation (6):

$$CeO_2 \rightarrow (Ce^{4+})_{1-2x}(Ce^{3+})_{2x}O_{2-x} + x/2\ O_2(g)\ (0 < x < 0.5) \tag{6}$$

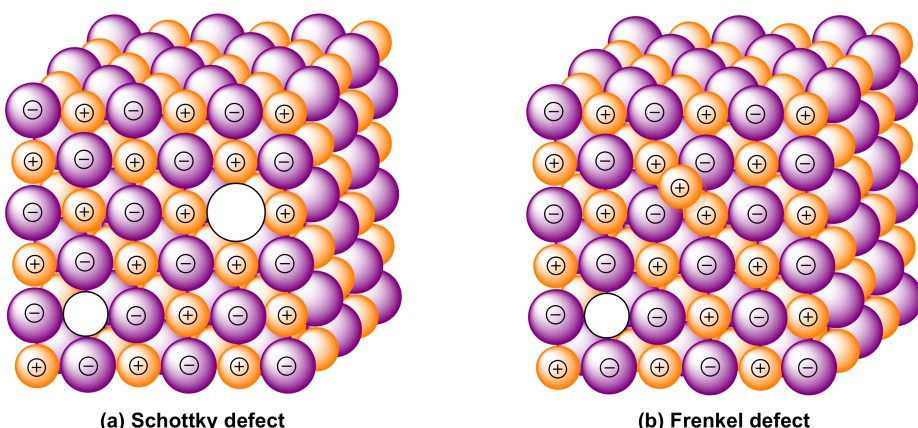

**(a) Schottky defect**         **(b) Frenkel defect**

**Figure 3.** Differences between (**a**) a Schottky defect and (**b**) a Frenkel defect in a lattice. Adapted from [132].

This defect forming process can also be written using the Kröger-Vink notations as shown in Equation (7):

$$CeO_2 \rightarrow 2xCe'_{Ce} + (1 - 2x)Ce_{Ce} + xV_O^{\bullet\bullet} + (2 - x)O_O + 0.5xO_2(g) \tag{7}$$

In this reaction, x moles of atomic oxygen are removed from the $CeO_2$ lattice, leaving oxygen vacancies and $(2 - x)O^{2-}$ anions at their original sites. To maintain electroneutrality, 2x moles of $Ce^{3+}$ ($Ce'_{Ce}$) will form, and the remaining $1 - 2x$ moles of Ce cations maintain the $Ce^{4+}$ ($Ce_{Ce}$) state. In undoped ceria, the concentration of oxygen vacancy defects $[V_O^{\bullet\bullet}]$ (only valid for very low concentrations) is proportional to $P(O_2)^{-1/6}$, where $P_{O2}$ is the partial pressure [136].

Oxygen vacancy defects of ceria enable ceria to act as an oxygen buffer to store oxygen in an oxygen-rich environment and release oxygen in an oxygen-deficient environment. This unique property is due to the interchange of the oxidation states $Ce^{4+}$ and $Ce^{3+}$ coupled with oxygen vacancy defects formation. This property is also called the oxygen storage capacity, which has been proven to be positively correlated with the activities of catalysts/electrolytes in automobile emission treatment systems and solid oxide fuel cells [107,109,137].

*2.3. Morphology*

The cubic fluorite $CeO_2$ structure possesses three low-index planes: (100), (110) and (111), depicted in Figure 4. The (100) planes consist of alternating charged planes that introduce a dipole moment perpendicular to the surface. These surfaces are not stable; however, they could be stabilized by defects or by the charge-compensating species present, for example, ligands or surfactants. The (110) surfaces are charge neutral, with stoichiometric proportions of anions and cations in each plane, which results in no dipole moment perpendicular to the surface. The (111) surfaces also exhibit no dipole moment perpendicular to the surface. Unlike the (110) planes, (111) surfaces consist of a neutral three-plane repeating unit terminated with a single anion plane [133].

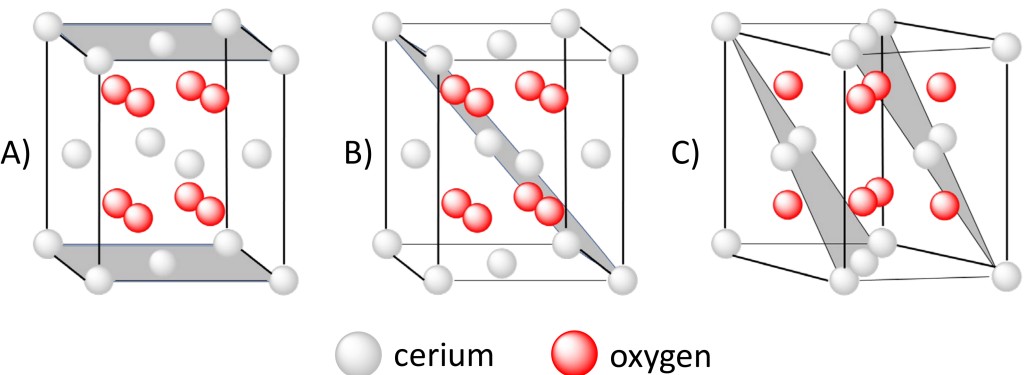

**Figure 4.** Different families of $CeO_2$ planes: (**A**) (100), (**B**) (110) and (**C**) (111). Adapted from [133].

The (111) plane is calculated to be the most stable facet irrespective of the different potentials used in the simulation, both before and after relaxation, according to the work done by Vyas [133]. The (110) plane is the next most stable facet, with a surface energy of 1.5 eV calculated from the Butler potential, while the (100) facet exhibits a surface energy of 2.0 eV, the highest among these three low-index facets [133]: *111 > 110 > 100* on the three low-index surfaces of $CeO_2$.

The most stable surface structure is the stoichiometric $CeO_2$ (111) surface under oxygen-rich conditions, with a surface free energy of 0.060 eV/Å$^2$ determined using the "ab initio atomistic thermodynamics" approach. In a reducing environment, the (111) surface with subsurface oxygen vacancies has been found to be the most stable surface, with a surface free energy of $-0.001$ eV/Å$^2$. In a highly reducing environment, a Ce-terminated (111) surface is the most stable [138]. The $CeO_2$ (110) surface with surface oxygen vacancies has a 0.012 eV/Å$^2$ surface free energy, which is 0.006 eV higher than that of the $CeO_2$ (111) surface with the same oxygen vacancies. The surface free energies of the $CeO_2$ (100) surface with the same type and amount of surface oxygen vacancies are 0.575 and 0.016 eV/Å$^2$ for surfaces terminated with oxygen and cerium, respectively, which are both larger than those of the $CeO_2$ (111) and $CeO_2$ (110) surfaces [138].

Thus, although the calculation methods used to evaluate the ceria oxide surfaces with similar surface structures (e.g., defects) are different, the $CeO_2$ (111) surface remains the most stable surface, followed by the (110) and (100) surfaces.

Other crystal planes of cerium oxides, such as the (200), (220), and (331), have also been investigated and characterized in both experimental and simulation studies [139]. For example, (220) facets were found in slightly truncated cerium oxide nano-cubes with predominant (100) facets synthesized by Kaneko et al. [140]. Sayle et al. reported surface energies of 11.577 and 2.475 J/m$^2$ for the (331) planes before and after relaxation by applying the energy minimization code Modular Interactive Acquisition System (MIDAS) [139]. Moreover, recent advances in materials science have improved the feasibility of tailoring the metal oxide morphology, and the desired crystal facets of cerium oxide materials can be preferentially exposed through precise control of the growth kinetics [98]. However, these three low-index planes are the most commonly observed and the most studied facets

of synthesized cerium oxide structures. They representatively illustrate the facet functions in different applications of cerium oxides.

## 3. Types of Mechanisms for Total Oxidation

The Mars and van Krevelen (MvK) mechanism was proposed, in 1954, to explain the kinetics of selective oxidation reactions over vanadium oxide catalysts [141]. In this mechanism, the hydrocarbon is oxidized by the metal cation with concomitant insertion of the surface oxygen leading to $CO_2$, $H_2O$ and an oxygen vacancy (rate of this step: $r_C$; kinetic constant: $k_C$) while the oxygen vacancy is refilled by gaseous $O_2$ (rate: $r_O$; kinetic constant: $k_O$). The steady state is reached for $r_C = r_O$, which leads to the rate equation:

$$r_{\text{MkV}} = \frac{k_C P_C \cdot k_O P_O}{v k_C P_C + k_O P_O)}$$

(8)

where $v$ is the overall stoichiometry of the reaction (one molecule of HC requires $v$ moles of $O_2$). This mechanism implies [142] that (1) both oxidation step and reoxidation of the surface are of first order with respect to HC and $O_2$, respectively, (2) O surface sites are free of carbon intermediates. In the literature specific to perovskite-based catalysts, oxidation mechanisms are often presented as "supra-facial" or "intra-facial" mechanisms [143–146]. Supra-facial mechanisms mean that only surface oxygens are involved in the catalytic reaction: they correspond to Langmuir-Hinshelwood or Eley-Rideal classical mechanisms, but also to the MvK mechanism in which the steps of O abstraction and O vacancy refilling are restricted to surface atoms. In intra-facial mechanisms, bulk oxygens are involved in the oxidation reaction. This is typically the case for MvK mechanisms, with participation of all the oxygens of the oxide network.

In the Eley-Rideal (ER) mechanism, $O_2$ is too weakly adsorbed so that it reacts directly from the gas phase with the adsorbed hydrocarbon molecule. The kinetic derivation of this mechanism leads to the following rate equation:

$$r_{\text{ER}} = k_C \frac{K_C P_C \cdot P_O}{(1 + K_C P_C)}$$

(9)

The kinetic order is +1 with respect to oxygen and between 0 (HC strongly adsorbed) and −1 (HC weakly adsorbed) for the hydrocarbon. However, the reverse can occur when gas phase HC or CO reacts with adsorbed O:

$$r_{\text{ER}} = k_O \frac{K_O P_O \cdot P_C}{(1 + K_O P_O)}$$

(10)

In the Langmuir-Hinshelwood (LH) mechanism, both hydrocarbon and oxygen are chemisorbed on the catalyst's surface and react according to a bimolecular reaction [147–149]. If the hydrocarbon and oxygen compete for the same sites, the kinetic derivation of the LH mechanism leads to the classical rate equation:

$$r_{\text{LH}} = k_C \frac{K_C P_C \cdot K_O P_O}{(1 + K_C P_C + K_O P_O)^2}$$

(11)

where r is the rate of HC consumption, $K_C$, the kinetic constant, $P_C$ and $P_O$, the partial pressure of HC and oxygen, and KC and KO their respective adsorption constants.

Depending on the relative adsorption strength of HC or $O_2$, kinetic orders may vary from −1 for the most strongly adsorbed reactant to +1 for the reactant that is weakly adsorbed. If the hydrocarbon and oxygen are adsorbed on distinct sites, the equation becomes:

$$r_{\text{LH}} = k_C \frac{K_C P_C \cdot K_O P_O}{(1 + K_C P_C)(1 + K_O P_O)}$$

(12)

Positive orders are then observed for all the reactants.

## 4. Total Oxidation of Methane on Ceria-Containing Transition Metal Oxide-Based Catalysts

As more lean methane is discharged into the environment with the increasing exploitation of natural gas fuel in many economic and industrial sectors, problems regarding environmental and health issues have led to the necessity for a solution to mitigate the methane and other volatile organic compounds accumulation in the atmosphere.

Researchers have directed their efforts towards the development of catalysts with high activity in the process of catalytic combustion of methane in order to solve this problem. Noble metal catalysts display excellent catalytic properties for combustion; however they are expensive and unstable at high operating temperatures, making their use difficult on a large scale. Transition metal oxide-based catalysts as non-noble metal catalysts represent an attractive alternative to replace the noble metal catalysts as they are abundant, cheap, more stable at high temperature, still less active in combustion than the noble metal ones, and nevertheless offer a valid option for the design of efficient catalysts for lean methane combustion. Among them, ceria-based catalysts have shown interesting catalytic activity in combustion processes, owing to their reducibility properties and oxygen storage capacity which is strongly related to morphological properties [150].

A study was performed by Li et al. to evidence the relationship between the $CeO_2$ structure/morphology and its catalytic performance in lean methane catalytic combustion [151]. They prepared sheaf-like and layered ceria via a hydrothermal method. The use of different solvents and different amounts of urea allowed the formation of two distinct ceria catalysts in regard to the final morphology: ethanol with 0.6 mol of urea assured the generation of sheaf-like $CeO_2$ structure, called $CeO_2$-S, and ethylene glycol with 0.1 mol of urea afforded the layered structure, noted as $CeO_2$-L. The Scanning Electron Microscopy (SEM) images for the as-prepared ceria catalysts are quite revealing, depicting for the sheaf-like structure a wheat bundle of nanofilaments tied up in the middle and spread loosely at both ends (Figure 5a,b), wherein, for the layered ceria catalyst, there are grouped multiple thin sheets of oxide in a low-range ordered structure (Figure 5c,d). Textural characteristics are relatively close to each other as Brunauer-Emmet-Teller (BET) surface area of $CeO_2$-S and $CeO_2$-L are 72.6 m$^2$/g and 68.2 m$^2$/g, respectively. The average crystal size for the sheaf-like ceria is 23.8 nm and for the layered ceria 24.9 nm, in accordance with the variation of their surface area.

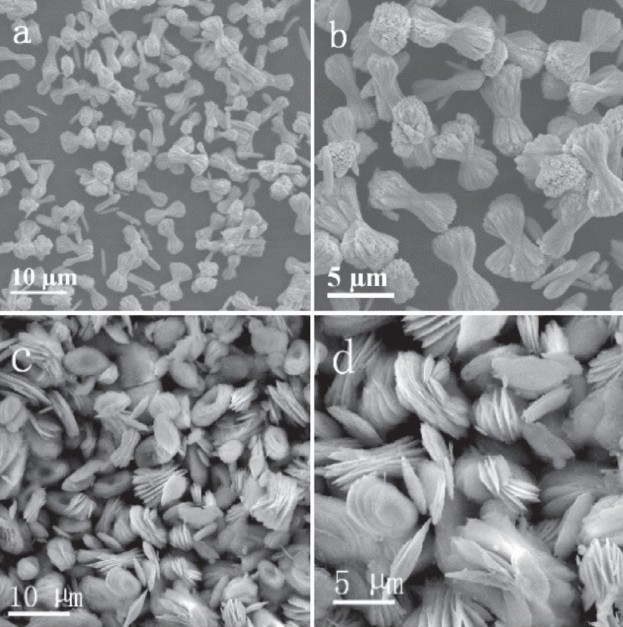

**Figure 5.** SEM images of $CeO_2$-S (**a**,**b**) and $CeO_2$-L (**c**,**d**). Reprint from [151]. Copyright (2020), with permission from Chemical Society of Japan.

The $CeO_2$ catalysts were tested in lean methane combustion, both indicating a good selectivity towards complete oxidation of methane, as the only reaction products were $CO_2$ and $H_2O$, with no CO being detected. $CeO_2$-S exhibits higher activity than $CeO_2$-L with methane conversion at 550 °C reaching 97% for the former and 94% for the latter (Figure 6).

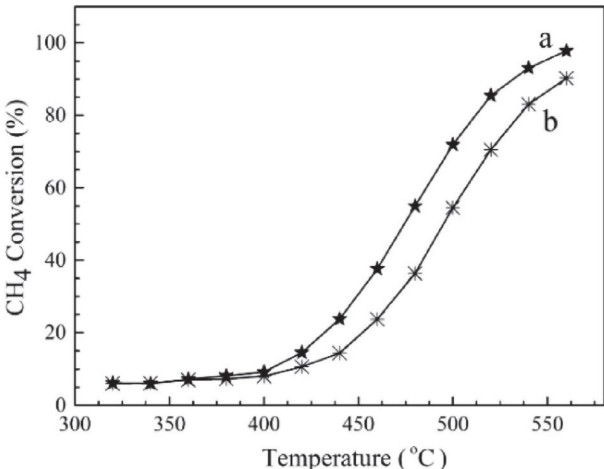

**Figure 6.** Catalytic activities in methane combustion of $CeO_2$-S (**a**) and $CeO_2$-L (**b**). Reprint from [151]. Copyright (2020), with permission from Chemical Society of Japan.

The difference in activity is attributed to the better reducibility of $CeO_2$-S, associated with the redox efficiency for changes in oxidation state between $Ce^{3+}$ and $Ce^{4+}$, facilitating the formation of oxygen vacancies and the migration of lattice oxygen, which is documented to have a beneficial effect on the $CH_4$ catalytic combustion by ceria-based materials [151]. $H_2$-TPR measurements support these results as the reduction peak appears at around 460 °C for $CeO_2$-S, while for $CeO_2$-L, it is located at around 500 C. Moreover, XPS experiments revealed a higher content of $Ce^{3+}$ at the surface level in $CeO_2$-S (16.23%) than that in $CeO_2$-L (14.36%), which corroborated the effect of $Ce^{3+}$ on the oxygen vacancies in $CeO_2$, implying that $CeO_2$-S is more susceptible to generate oxygen vacancies compared with $CeO_2$-L, with great impact upon the efficiency in the lean $CH_4$ catalytic combustion. In addition, XPS measurements showed a higher content of adsorbed oxygen (active oxygen) for the sheaf-like catalysts than that of the layered one. Therefore, in this study, the sheaf-like $CeO_2$ proved to be more active in methane combustion, thanks to its unique morphology which induces better reducibility and a higher number of oxygen vacancies.

Furthermore, Sugiyama's group has evaluated the methane combustion performance of several ceria catalysts with different specific surface areas, obtained through a thermal hydrolysis route [152]. The use of ceric nitrate, $(H_x[Ce(NO_3)_{4+x}])$, instead of cerous nitrate, $Ce(NO_3)_3$, after precipitation with ammonia and calcination at 400 °C for 10 h, afforded a type of high thermal resistant ceria with higher surface area at 900 °C (50 $m^2$/g) and smaller particle size (34 nm) than its counterpart, which does not retain its textural characteristics, with Surface Specific Area (SSA) of 4.1 $m^2$/g and particle size more than 100 nm. The high thermal resistant ceria has an initial particle size of 9 nm after calcination at 400 °C and maintains its relative scale, with a low increase in particle size towards 34 nm even after treatment at high temperature, indicating its excellent thermal properties, as depicted in TEM images (Figure 7). Moreover, this study revealed a higher and more stable activity in methane combustion for the thermally resistant ceria compared to the less resistant reference, this fact being ascribed to its larger pore volume and high mesoporous structure, with mean pore size of around 7 nm, maintained even after calcination at 900 °C, indicating limited sintering of the ceria particles during calcination which results in the desired specific surface area. However, the tested catalysts were not capable of achieving the complete oxidation of methane in the temperature range from 500 to 650 °C as more CO was observed during the combustion process.

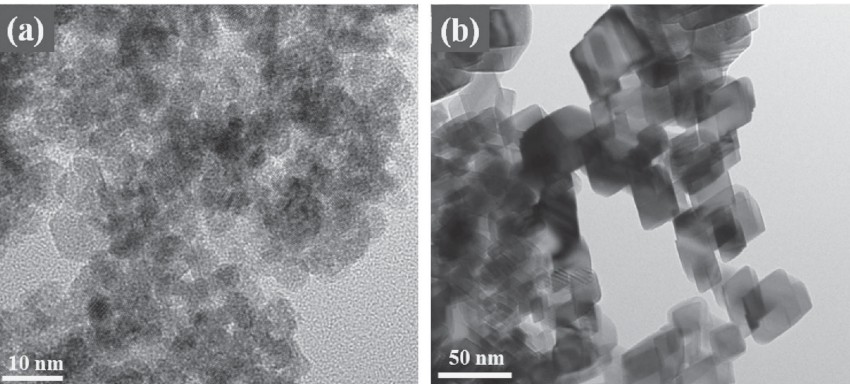

**Figure 7.** TEM image of high thermal resistant ceria after calcination at 400 °C (**a**) and 900 °C (**b**) under air. Reprint from [152].

Nevertheless, the role of high specific surface area in ceria-based catalysts was ascertained, revealing a direct correspondence with the oxygen storage capacity in catalysts, which is an important factor to be considered in catalytic combustion activity.

### 4.1. Monometallic Modified Ceria Catalysts

More attempts to develop ceria-based catalysts have been realized with the purpose of improving the catalytic activity of ceria in methane combustion. Such attempts have consisted of doping $CeO_2$ with metal ions with a lower oxidation state than 4+ in order to create more oxygen vacancies in the ceria structure, as the choice of dopants can control the properties of these oxygen vacancies (local atomic structure, chemical environment, atom binding energies).

Work [153] regarding the design of ceria catalysts was conducted by preparing samples of doped $CeO_2$ with an alkaline earth metal (Ca), rare earth metal (Nd), a transition metal (Mn) and a p-block metal (Pb), then comparing their catalytic activities in methane combustion to that of an undoped $CeO_2$ sample [153]. The oxide catalysts were prepared via a hydrothermal method where solution of metal nitrates was treated with 0.2 M ammonium oxalate solution so that the metal oxalates coprecipitate, after which they were calcined for 4.5 h at 600 °C in air.

Different concentrations of doping element were achieved, ranging from 2% in the case of Mn (Mn2), 10% for Ca (Ca10), 11% for Pb (Pb11), and 11% and 31% for Nd (Nd11, Nd31) to describe the effect of the dopant on the catalytic activity. All the prepared samples presented average particle sizes after calcination of between 15 and 20 nm with specific surface area of 29–43 $m^2/g$.

The catalytic activity of the doped ceria catalysts was determined using a gas mixture of 1.4% $CH_4$, 25% $O_2$ in He, in the temperature range 400–620 °C where complete oxidation of methane was obtained, as the products of an incomplete oxidation (CO, $H_2$) were not detected (Figure 8). Notable differences in activity were observed for the tested samples, with the Nd11 sample displaying the highest activity in combustion as evidenced by its lowest $T_{20}$ and $T_{50}$ temperatures of 525 and 579 °C, respectively, which correspond to a 20% and 50% conversion of methane.

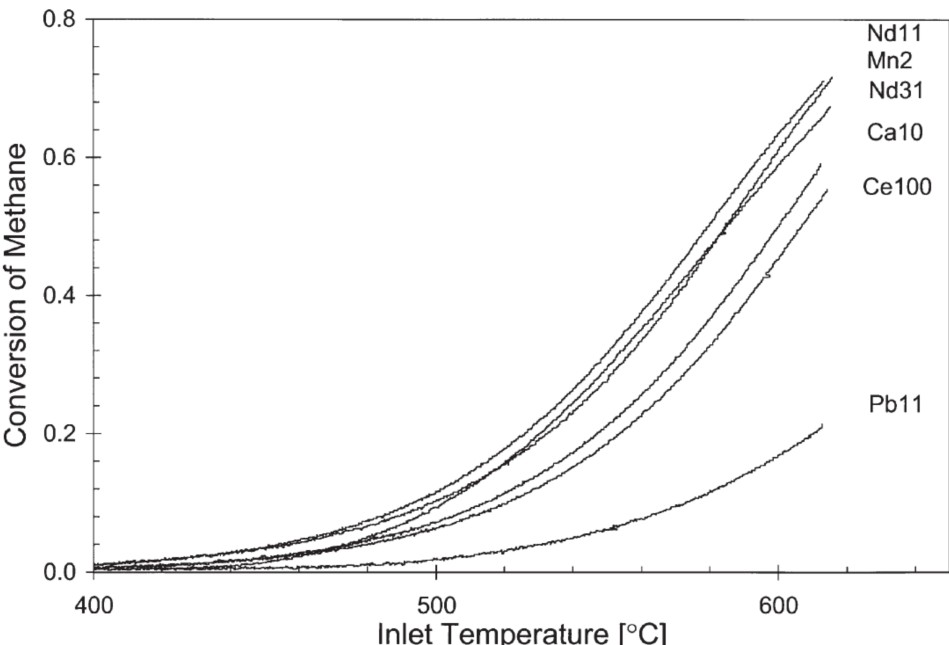

**Figure 8.** The conversion of methane versus temperature for the doped ceria catalysts. Reprint from [153]. Copyright (196), with permission from Springer.

The activity of the Mn2 sample followed closely that of the Nd11 sample, showing high catalytic performance with $T_{20}$ of 530 °C and $T_{50}$ of 585 °C. Except for Pb-doped ceria, all the doped catalysts exhibited improved catalytic activity compared to the undoped ceria with $T_{20}$ of 553 °C and $T_{50}$ of 607 °C.

The effect of dopant elements with lower oxidation states than 4+ as they are incorporated in the $CeO_2$ structure consists of the formation of oxygen vacancies and local structure changes, while the coordinatively unsaturated surface Ce ions content increases. In the case of cerium oxide, the cerium ion has two oxidation states which are crucial in redox reactions, generating oxygen vacancies and further coordinative unsaturation at the surface which cause underlying bound oxygen ions to be weaker than those of the bulk. These weakly bound oxygen ions are more active and participate in oxidation reactions [98,153].

Moreover, as lower-valency dopants introduce more oxygen vacancies in the ceria structure, providing more active sites for the oxygen transfer, the mobility of the oxygen ions in the lattice also influences the overall catalytic activity. The Nd- and Ca-doped ceria catalysts displayed higher ionic conductivities than the undoped $CeO_2$, indicating a relationship between the oxygen ion mobility and the catalytic activity in methane combustion [154]. This study showed a successful attempt at catalytic activity improvement through metal ion doping.

Research focused on copper as doping metal in ceria catalysts involved the preparation of $CuO/CeO_2$ catalysts through the impregnation method to show the influence of the support on the active phase in methane combustion. Four different methods (citric acid sol-gel method, hydrothermal method, thermal decomposition of $Ce(NO_3)_3 \cdot 6H_2O$ and $Ce(NH_4)_2(NO_3)_6$) afforded different ceria supports denoted as $CeO_2(s)$, $CeO_2(h)$, $CeO_2(t1)$ and $CeO_2(t2)$, respectively, to be used together with commercial ceria, $CeO_2(c)$, in order to obtain $CuO/CeO_2$ catalysts via wet impregnation methods with $Cu(NO_3)_2$ aqueous solution [155]. After drying overnight at 110 C, the impregnated materials were calcined at 550 C for 4 h in air to achieve a final Cu loading of 5 wt%.

A comparison of the textural properties reveals that the BET surface area of the Cu-doped catalysts varies considerably from 68.7 $m^2/g$ for the $CuO/CeO_2(t1)$ sample to only 0.7 $m^2/g$ for the Cu-doped commercial ceria. Regarding the crystallite size of $CeO_2$, in accordance with the BET results, the $CuO/CeO_2(t1)$ has the smallest particle size among

the catalysts (11.9 nm), whereas the $CuO/CeO_2$(c) presents the largest particle size in the series (49.8 nm).

The crystallite size of ceria, alongside the weakest reflections of CuO in the X-ray diffraction (XRD) patterns which can be ascertained by the corresponding size of the CuO particle, indicate the highest dispersion of copper in the $CuO/CeO_2$(t1) catalyst, which plays an important role in catalytic activity. The SEM images of $CuO/CeO_2$ catalysts showed that the preparation method of ceria supports greatly affects the particle size and the morphologies of the catalysts, as confirmed by XRD and BET results (Figure 9).

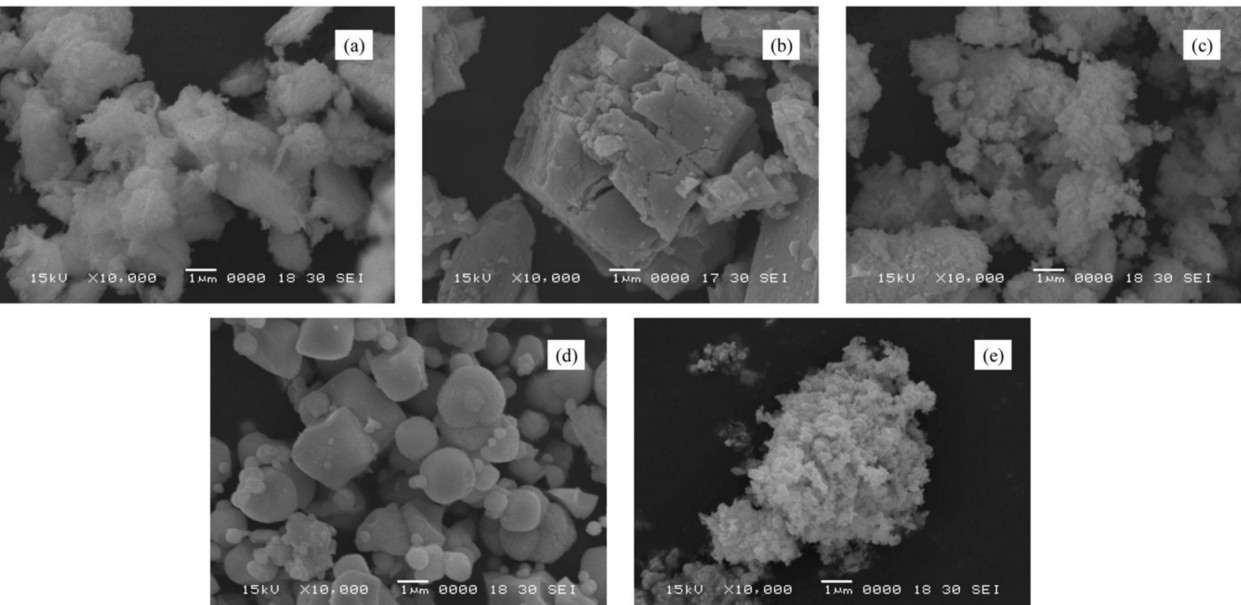

**Figure 9.** SEM images of different Cu-doped ceria catalysts: (**a**) $CuO/CeO_2$(t1); (**b**) $CuO/CeO_2$(c); (**c**) $CuO/CeO_2$(s); (**d**) $CuO/CeO_2$(h); (**e**) $CuO/CeO_2$(t2). Reprint from [155]. Copyright (2009), with permission from Elsevier.

The as-prepared catalysts were tested in methane conversion in the temperature range 300–650 °C, achieving complete oxidation of the hydrocarbon with $CO_2$ and $H_2O$ as the only products (Figure 10). Except for $CuO/CeO_2$(c), Cu-doped catalyst series exhibited higher activity than pure CuO and, among them, $CuO/CeO_2$(t$_1$) had the highest catalytic activity which decreases in the following order: $CuO/CeO_2$(t$_1$) > $CuO/CeO_2$(s) > $CuO/CeO_2$(h) > $CuO/CeO_2$(t$_2$) > $CuO/CeO_2$(c).

These results suggest a synergetic interaction between CuO and ceria support, which leads to a strong metal oxide-support interaction as the reason for enhanced catalytic activity in methane combustion. In accordance with the textural properties of the different prepared $CeO_2$ supports, different dispersion of CuO was achieved in catalysts, subsequently rendering their activity in direct relation to their reducibility. Temperature-programmed reduction (TPR) measurements revealed that Cu-doped commercial ceria was the least active catalyst in the series with the highest reduction temperature, followed further by $CuO/CeO_2$(t2), $CuO/CeO_2$(h), $CuO/CeO_2$(s) and $CuO/CeO_2$(t1) as their reduction temperature decreased. Therefore, $CuO/CeO_2$(t1) catalyst showed the highest activity with a low temperature peak at 140 °C in TPR. The high dispersion of CuO particles allowed more $Cu^{2+}/Cu^+$ redox couples which greatly impact the activity in oxidation reaction, and as a consequence the increase of ceria particle size favors less associated CuO particles with the support, manifesting in more inactive bulk-like CuO doped catalysts.

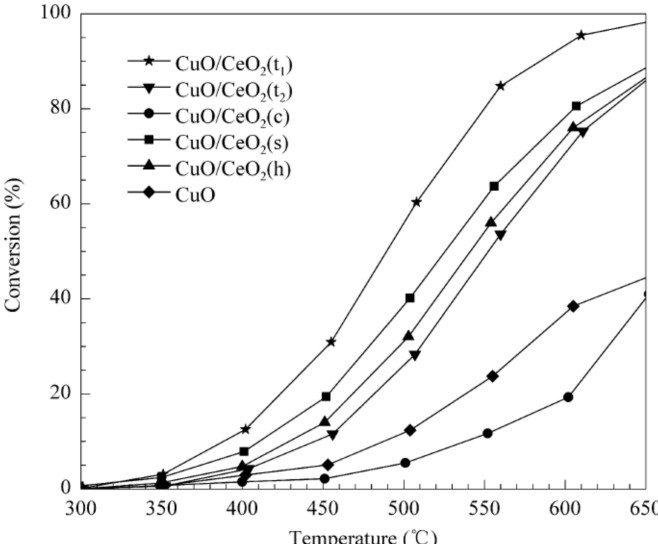

**Figure 10.** Catalytic activities in methane conversion for different Cu-doped ceria catalysts. Reprint from [155]. Copyright (2009), with permission from Elsevier.

This study also evidenced the effect of Cu loading on the best performing ceria support (CeO$_2$ obtained from cerous nitrate, Ce(NO$_3$)$_3$·6H$_2$O, thermal decomposition), CeO$_2$(t1), on the efficiency in methane combustion (Figure 11). The catalytic activity increased naturally with Cu loading on the support until it reached 5 wt%; after this point, the catalysts with 10% and 15% Cu showed similar activities to that of 5 wt%. This behavior suggested that a maximum dispersion of CuO particles onto the support surface is reached at 5 wt% with no more beneficial effect on the catalytic activity as the Cu loading increased, indicating a saturation of the surface in the active phase. Hence, at larger Cu content, a layer of bulk-like CuO formed besides the existing highly dispersed monolayer, which provides the maximum catalytic activity.

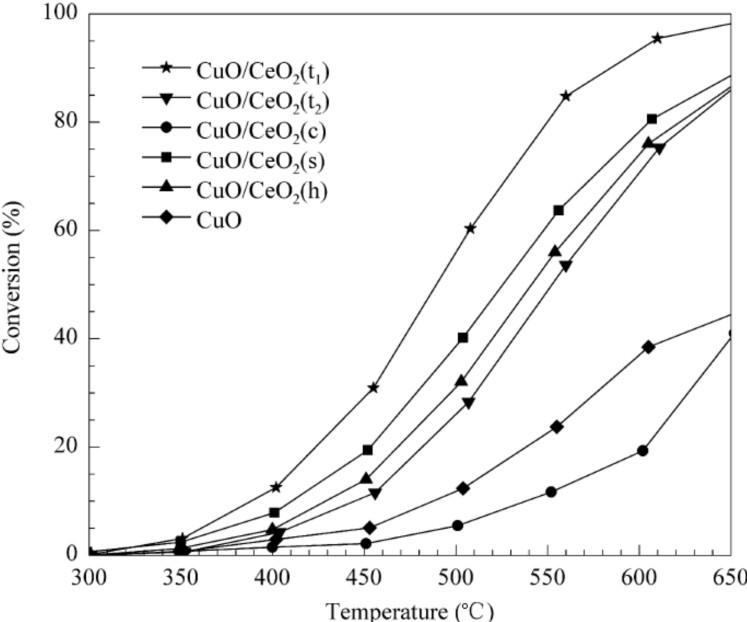

**Figure 11.** Catalytic activities of Cu-doped ceria catalysts in methane conversion for different Cu loading. Reprint from [155]. Copyright (2009), with permission from Elsevier.

A different strategy to prepare Cu-doped ceria catalysts consists of using an aerosol-based two-stage thermal treatment method to produce a Cu/Ce-O hybrid nanoparticle (NP) with a tunable oxidation state via hydrogen reduction process, as depicted in Figure 12 [156].

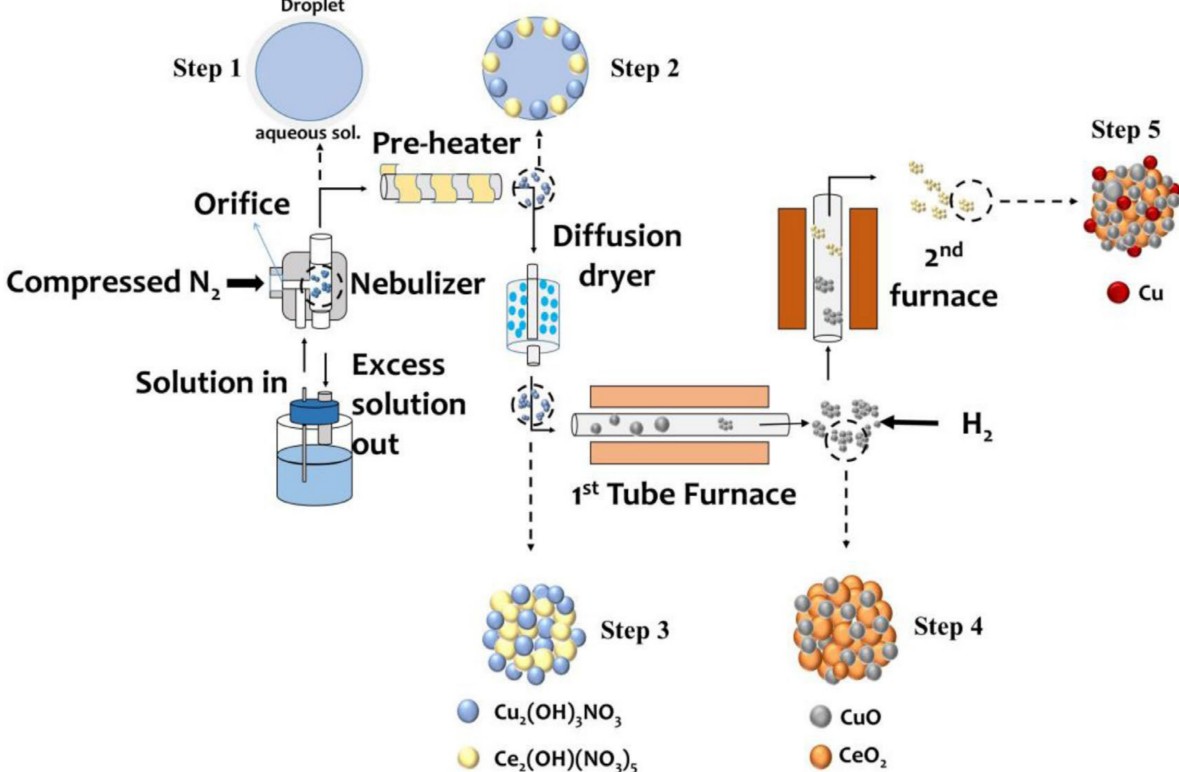

**Figure 12.** Scheme of the two-stage thermal treatment to synthesize Cu–Ce–O NPs with tunable oxidation state. Reprint from [156]. Copyright (2018), with permission from Elsevier.

Production of aerosol particles allowed a homogenous distribution of the elements in the hybrid nanostructure to generate an extended Cu-Ce-O interface, providing a scaffold where Cu presents different oxidation states ($Cu^0$, $Cu^+$ and $Cu^{2+}$), with strong metal-support interactions for improving the catalytic activity. A peculiar approach in the element composition for the samples focuses on the low content of Ce (1.9 wt%) and Cu (5 wt%), with different thermal treatments (300, 400, 500 and 600 °C) of the hybrid nanoparticles to achieve diameter size of 94–98 nm. XRD measurements for the $CuCeO_x$-NP-500 and $CuCeO_x$-NP-600 samples have shown multiple phases corresponding to the Cu crystalline, with crystallite sized between 10 and 20 nm, the crystallinity of Ce and its oxidation state remaining unchanged at different decomposition temperatures. The XPS results confirmed the XRD analysis, indicating the multiple oxidation states of Cu without changing the cerium oxidation state in the hybrid nanoparticles using the aerosol based strategy. Moreover, the surface fraction of Cu increased with the decomposition temperature, suggesting a more efficient reduction of CuO with $H_2$ at higher temperature as Cu started to aggregate on the nanoparticle surface.

The evaluation of the catalytic activity of this series was performed by methane combustion of a gas mixture of 11% $CH_4$ and 89% air at an operating temperature ranging from 200 °C to 500 °C. The tested catalysts showed an impressive light-off temperature at around 340–360 °C, evidencing the strong metal-support interaction present at the interface of $CuO_x$ and $CeO_2$ (Figure 13).

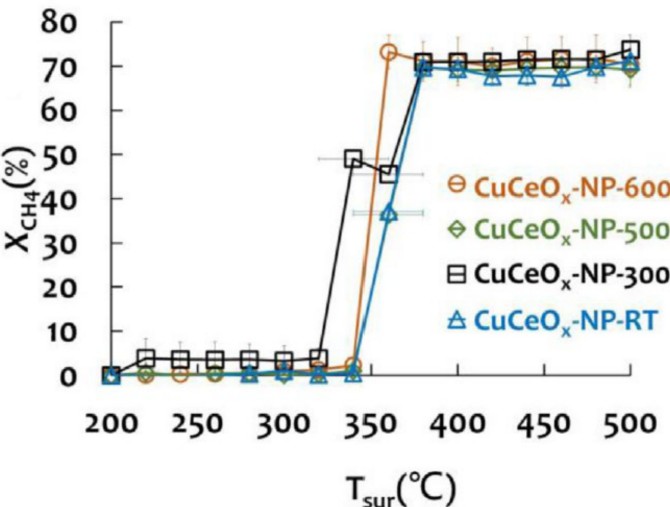

**Figure 13.** Conversion of methane over Cu-Ce-O NPs versus temperature. Reprint from [156]. Copyright (2018), with permission from Elsevier.

These results highlighted the capability of $CeO_2$ to generate oxygen vacancies via $Ce^{4+}/Ce^{3+}$ redox couples at the Cu–Ce–O interface and, hence, to adsorb oxygen from the air, while the $CuO_x$ phases provide more active sites for methane adsorption, as no more active sites on Cu crystallites are used for the initial oxygen adsorption, managing the adsorption competition between methane and oxygen. Nevertheless, the final results showed that the catalytic performance is not affected by the initial oxidation states of Cu in the hybrid interface of NPs [156].

At the same time, the synergetic effect of $CuO_x$ and $CeO_2$ consists in the improvement of the light-off temperature and an increased stability in combustion processes, compared with only $CuO_x$-NPs discussed in this work (Figure 14). The stability in methane combustion over the prepared NP catalysts was evaluated at a reaction temperature of 500 °C. $CuCeO_x$-NPs showed a lower decreasing rate in conversion during the 8 h test period than the $CuO_x$-NPs, varying from 70% to 50%, in comparison to 70% to 30% for the latter [156].

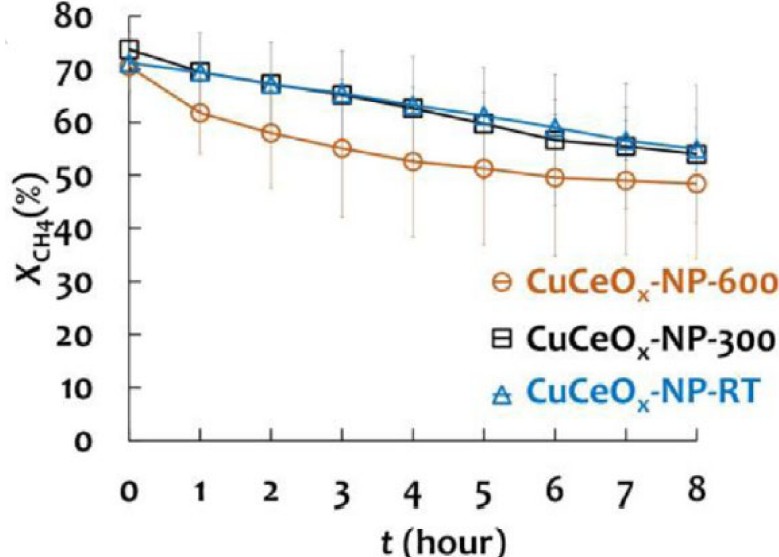

**Figure 14.** Stability tests of the catalytic activity of $CuCeO_x$-NPs. Reprint from [156]. Copyright (2018), with permission from Elsevier.

The problem of reusability was treated by XRD analysis of the used catalysts in methane combustion, as no more crystalline patterns for Cu and $Cu_2O$ were observed,

except those for CuO and CeO$_2$, indicating a continuous oxidation of Cu during the methane combustion. Furthermore, the calculation of particle size revealed an increase of CuO crystallite after the 8 h test, suggesting a sintering process during the combustion process. However, this work demonstrates the synergetic effect between CuO as active phase and ceria support to improve the catalytic activity in methane combustion, offering a novel synthesis procedure to manufacture different hybrid nanoparticles.

Cu-doped ceria catalysts were also the focus of other works where catalysts with different Cu loading ranging from 1.5 to 15 wt% were prepared through solution combustion synthesis (SCS). This method involves the preparation of aqueous solution containing metal salts (cerium and copper nitrate) and citric acid in excess to provide the fuel-rich conditions required for the solution combustion process [157,158]. Firstly, the solution was heated at 90 °C to form a gel which was then treated at 380 °C to obtain a fluffy sponge-like solid material after combustion. Finally, the solid was calcined for 4 h at 550 °C in air to afford the Cu-doped catalysts characterized by a spongy morphology with a macro porous coral reef-like scaffold, which is well evidenced for the pure ceria synthesized in the SCS method as depicted in Figure 15a. The gases released during the solution combustion synthesis introduced an advanced grade of porosity as large voids in a spongy-like structure in the case of 6% CuO-CeO$_2$ material (Figure 15c).

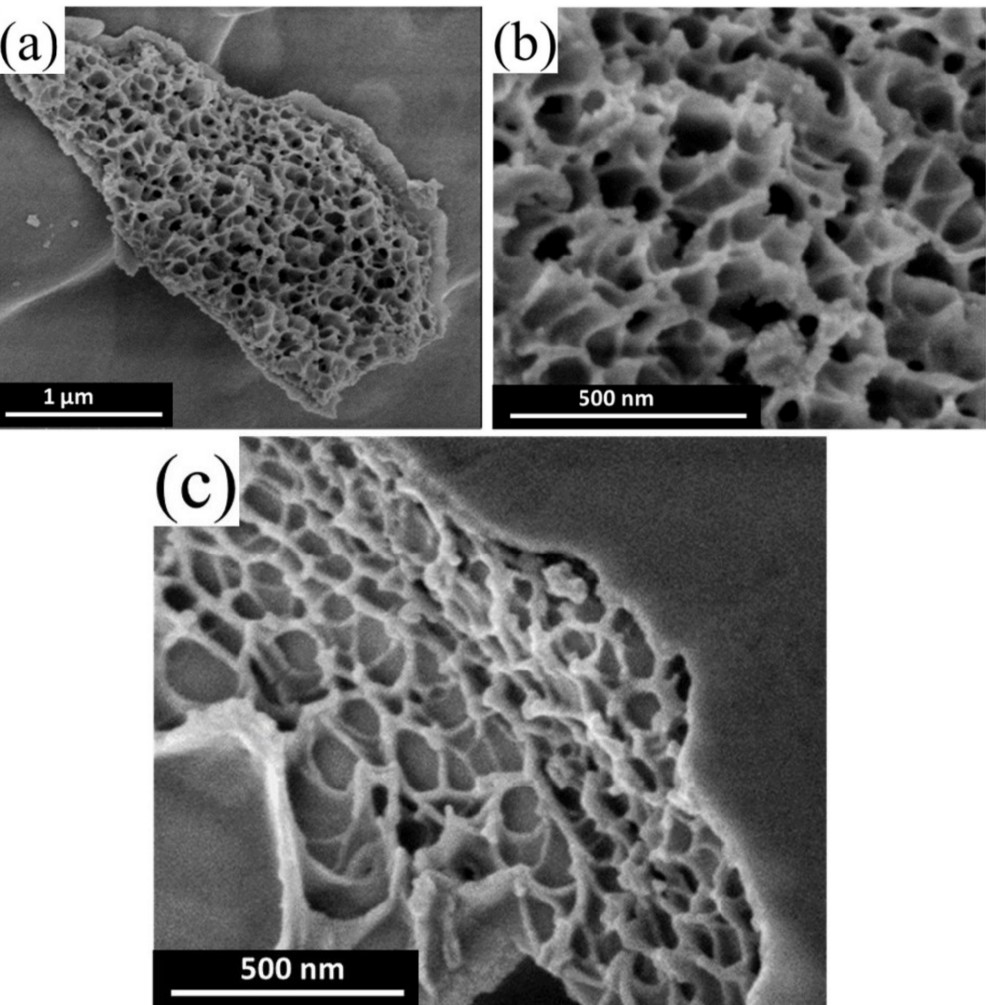

**Figure 15.** SEM images of CeO$_2$ (**a,b**) and 6% CuO-CeO$_2$ (**c**) prepared by solution combustion synthesis. Reprint from [157].

The XRD analysis revealed a reduction in average crystallite size of CeO$_2$ from around 19 nm in the case of pure CeO$_2$ to around 9 nm in the Cu-doped ceria material, in accordance

with the main XRD peak broadening for ceria oxide phase as the introduced copper was accommodated into the ceria lattice.

The catalytic performance in $CH_4$ combustion over $CuO-CeO_2$ materials was evaluated from the methane conversion versus catalyst temperature light-off curves, using a feed gas containing 100 ppmv $CH_4$, 20% (*v/v*) $O_2$ and Ar for the remainder. As expected, compared to the pure $CeO_2$ materials, the Cu-doped ceria catalysts showed enhanced activity in methane oxidation with complete combustion towards carbon dioxide and no carbon monoxide gas being detected. Among the tested catalysts, 6% $CuO-CeO_2$ exhibited the highest catalytic activity with a $CH_4$ conversion of 93% at 585 C, followed in the series by the 4.5% $CuO-CeO_2$ with a $CH_4$ conversion of 80% at the same temperature of 585 °C (Figure 16). Comparing the methane conversion at 500 °C and 585 °C, the activity of Cu-doped ceria catalysts decreases in the following order: 6% $CuO-CeO_2$ > 4.5% $CuO-CeO_2$ > 15% $CuO-CeO_2$ > 3% $CuO-CeO_2$ > 1.5% $CuO-CeO_2$. An enhanced activity is attributed to the improved oxygen mobility through the ceria oxide bulk and surface as more copper ions were inserted in the lattice.

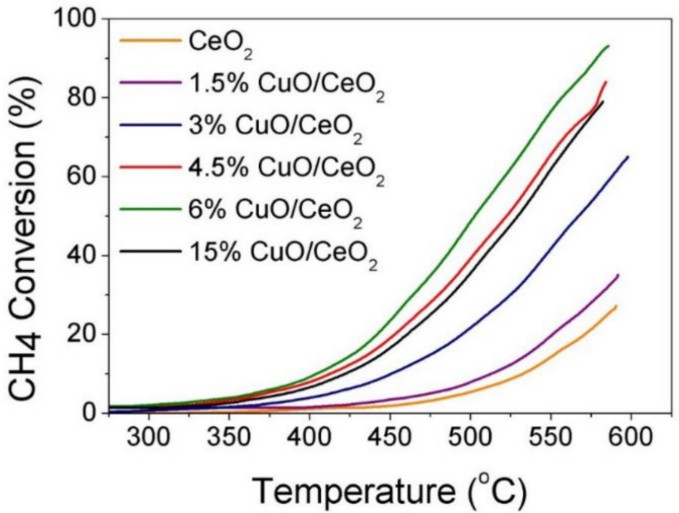

**Figure 16.** $CH_4$ conversion over different loaded $CuO-CeO_2$ materials versus temperature. Reprint from [157].

These results indicate a volcano-like dependence of Cu loading content over the catalytic activity as the 15% $CuO-CeO_2$ displayed a lower activity than the 6% Cu-doped material. This fact is ascribed to the formation of a separate phase of CuO at larger Cu content presenting weaker interactions with the ceria support. Nevertheless, this study provides additional evidence of the synergetic effect between the active phase of transition metal oxide and the ceria support, as the doped catalysts exceed the activity of pure $CeO_2$. Each oxide performs its function, where the metal oxide facilitates the dissociative adsorption of methane, while the oxide supports provide oxygen from the lattice, which is replenished by adsorption and dissociation of $O_2$ from the feed gas, in accordance with the Mars van Krevelen mechanism [158].

The preparation method via solution combustion synthesis provided a type of Cu-doped ceria catalyst, more active and capable to light-off and convert methane in very low concentration from feed gas, compared to the other discussed Cu-doped ceria materials whose crystalline phases of CuO and $CeO_2$ were more dissociated and segregated, their activity having been tested with a more methane concentrated gas mixture. Moreover, these results suggest that the insertion of transition metal ions in the ceria lattice improves considerably the catalytic performance of the materials in methane combustion as the oxygen ion mobility is enhanced and the generation of oxygen vacancies is promoted more.

Other research centered on the doping of $CeO_2$ with lanthanum since $La^{3+}$ ions are more compatible with the ceria lattice and can be accommodated more easily into the

structure with great effects on the textural and redox properties of the material, as the doping process could inhibit the sintering at high temperature and decrease the reduction temperature of $CeO_2$, improving its reducibility [159,160].

A series of Ce–La–O catalysts was prepared by sol-gel method using an aqueous solution of $La(NO_3)_3 \cdot 6H_2O$, $Ce(NO_3)_3 \cdot 6H_2O$ and citric acid to form a gel which was dried, then calcined at 550 °C for 4 h in air. The mixed oxides are discussed based on the different ratios between Ce and (Ce+La) which is defined as X in the notation Ce(X)–La–O [161]. XRD measurements describe the effect of $La^{3+}$ in the mixed oxide structure as the particle size of ceria decreases with the increase in $La^{3+}$ content from 11 nm for Ce(0.8)–La–O to 5 nm for Ce(0.2)–La–O. Moreover, XRD patterns specific to $CeO_2$ are only observed at samples with Ce/(Ce+La) ratios larger than 0.2. The estimation of BET surface area for the catalyst series shows that the mixed oxide samples present much larger surface areas than those of pure oxide, reaching a maximum of 67 $m^2/g$ in Ce(0.7)–La–O. This implies that the insertion of La into ceria promotes a more efficient dispersion of oxides; however, when the ratio of Ce/(Ce+La) is lower than 0.5, the surface area of materials drops considerably with the increase of La loading, indicating an equilibrium between the $La_2O_3$ and $CeO_2$ phases in matters of dispersion.

The catalytic activity of this series was evaluated by combustion tests of flow gas containing 0.2% $CH_4$ and 10% $O_2$ (balanced with $N_2$) where the reaction produced only $CO_2$ and $H_2O$ with no other by-products. As expected, the catalysts with the largest surface area exhibited the highest activity as the interplay between the surface active species (superoxide ions) and centers for methane adsorption are more numerous, with more surface area exposed by materials (Figure 17). Consequently, Ce(0.6)–La–O showed the highest catalytic performance in methane oxidation from the tested series, in accordance with the TPR results, where it consumed the highest amount of $H_2$ at lower temperature, and to the Raman spectra which indicated the most intense band at 1167 $cm^{-1}$ ascribed to the presence of superoxide ions [162]. In activity, it is followed by the other catalysts with relative low loading of La in the order: Ce(0.6)–La–O > Ce(0.7)–La–O > Ce(0.5)–La–O > Ce(0.8)–La–O > Ce(0.4)–La–O > Ce(0.3)–La–O > Ce(0.2)–La–O. The large amount of La inhibits the adsorption of active oxygen required for the combustion process since the available surface area is significantly lowered, even though the La basic species favors the activation of methane molecule by splitting of the C–H bond [163].

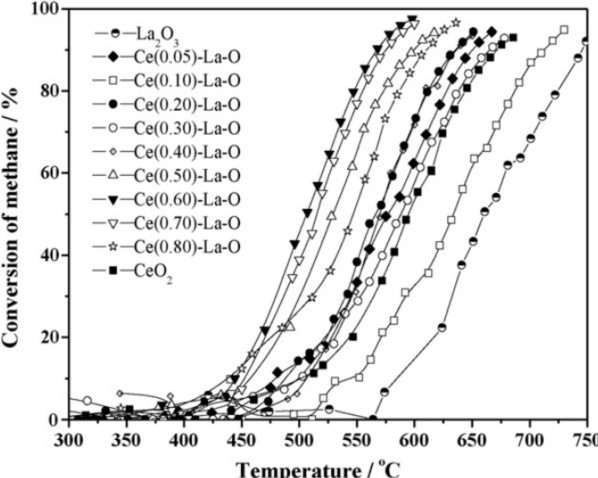

**Figure 17.** Catalytic activity of Ce–La–O catalysts in methane combustion. Reprint from [161]. Copyright (2010), with permission from Elsevier.

Doping of ceria materials with lanthanum ions proved to be successful since the obtained catalysts had smaller particle size, larger surface area and increased capacity for oxygen storage as La incorporation into $CeO_2$ lattice induces the formation of non-

stoichiometric $CeO_{2-x}$, which presents oxygen vacancies, hence displaying enhanced activity in methane combustion compared to the undoped ceria.

More attempts to improve the catalytic performance of ceria materials were made by doping $CeO_2$ with Zr with 10% mol via solution combustion synthesis, a previously mentioned method, where $\alpha$-amino-acids (glycine, lysine and alanine) were used as fuels in combination with the metal nitrates, manufacturing samples associated with the utilized amino-acid, denoted as GS, LS and AS, respectively, which were calcined at 600 °C in air [164]. The type of amino-acid produced multiple differences between the samples, depending on the length of hydrocarbon chain which relates to the volume of generated gas during the synthesis process. Textural characterization indicated that the GS-CeZrO sample presented the highest specific surface area (45 $m^2$/g) and pore volume (0.086 $cm^3$/g) with a particle size of ca. 9 nm, compared to LS and AS samples with a specific surface area of ca. 5 and 13 $m^2$/g, pore volume of 0.011 and 0.028 $cm^3$/g, and similar particle size (8 and 9 nm), respectively.

Catalytic tests performed using a feed gas with 2% $CH_4$, 4% $O_2$ and $N_2$ for balance revealed the highest activity in methane oxidation, as expected, for the GS sample thanks to its large surface area and pore volume, with $T_{50}$ lower at 100 °C than AS material (Figure 18).

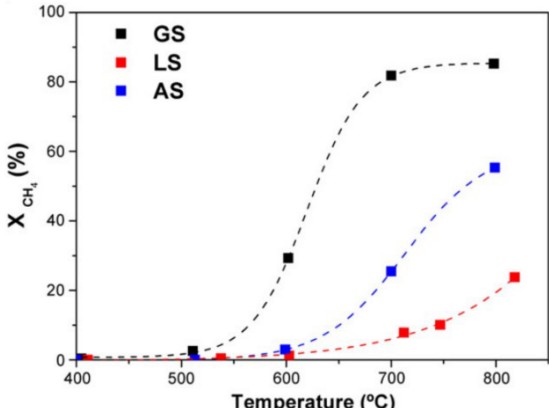

**Figure 18.** Methane conversion for the Zr-doped ceria catalysts. Reprint from [164]. Copyright (2011), with permission from Elsevier.

This behavior is ascribed to the higher content of $Ce^{4+}$ sites on the surface to adsorb oxygen from the feed gas, therefore the reduction in particle size implies more available surface area for the gas exchange processes, enhancing the catalytic performance. The observed reactivity is explained by the presence of atoms on the surface with certain geometric arrangements and electronic properties required for the interaction with the reactant molecules. Furthermore, these atoms are rendered less coordinated with diverse and increased reactivity, and at the same time, also exist in larger numbers as more specific surface area is exposed to the atmosphere. This study evidenced the crucial role of the surface area in influencing the catalytic activity in methane oxidation; however, this ZrCeO catalyst does not match the performance of the Cu- or La-doped ceria materials, also prepared through sol-gel methods.

Other transition metal doped catalysts were prepared by the co-precipitation method where a solution containing iron and cerium ammonium nitrates was treated with aqueous solution of 10% $NH_3$. The precipitates were calcined after drying at 500 °C in air for 2 h, and the afforded mixed oxides, $Ce_{1-x}Fe_xO_{2-\delta}$, presented doping metal concentration in the range of 5%–40% (x = 0.05–0.4) [165]. Unexpectedly, the XRD diffraction patterns of all the samples showed the peaks ascribed to the fluorite cubic $CeO_2$ structure, however with no observation of diffraction peaks for $Fe_2O_3$ phase, even in the more loaded Fe materials. Moreover, the main peak of $CeO_2$ shifts to higher diffraction angles, and also starts to broaden, suggesting an incorporation of Fe ions into the ceria lattice to form a Ce–Fe–O

solid solution which results in a contraction of the unit cell, as $Fe^{3+}$ (0.064 nm) is smaller than $Ce^{4+}$ (0.101 nm) [166].

Therefore, with the increase of Fe content in the samples, the diffraction peaks broaden more, resulting in smaller grain size as can be observed from the textural characteristics. As the Fe concentration increased in the ceria materials, the crystallite size decreased from 8.9 nm (pure $CeO_2$) to 2 nm for the most Fe loaded catalyst, $Ce_{0.6}Fe_{0.4}O_{2-\delta}$, and consequently the BET specific surface areas increased from 64 $m^2/g$ in pure ceria to 114 $m^2/g$ in $Ce_{0.6}Fe_{0.4}O_{2-\delta}$.

Furthermore, Raman spectra were able to detect free $Fe_2O_3$ particles on the most Fe loaded catalysts (x = 0.35 and 0.4), indicating the very small size and the high dispersion of these particles on the catalysts' surface, whereas XRD diffraction could not discern their presence. TPR measurements evidenced the superior reducibility of the $Ce_{0.65}Fe_{0.35}O_{2-\delta}$ and $Ce_{0.6}Fe_{0.4}O_{2-\delta}$ catalysts in contrast to the $Ce_{0.7}Fe_{0.3}O_{2-\delta}$ sample, as they contain, besides the mixed oxide solution, free $Fe_2O_3$ particles which provide more redox sites.

Combustion tests with a gas feed consisting of 1% $CH_4$, 20% $O_2$ and $N_2$ in balance were performed to investigate the activity of Fe-doped catalysts which revealed that the most loaded Fe sample, $Ce_{0.6}Fe_{0.4}O_{2-\delta}$, exhibits the highest catalytic activity in methane oxidation with $T_{10}$, $T_{50}$ and $T_{90}$ of 333, 378 and 438 °C, respectively, followed closely by the $Ce_{0.7}Fe_{0.3}O_{2-\delta}$ with a $T_{90}$ of 458 °C (Figure 19). This excellent performance of the highest Fe loaded sample is attributed to the co-existence of free $Fe_2O_3$ particles with a Ce–Fe–O solid solution where the active oxygen vacancies reside. In addition, the introduction of Fe ions in the ceria lattice had a great impact on the activation energy of the methane oxidation, dropping from around 100 kJ/mol for pure $CeO_2$ to 51.1 kJ/mol for the $Ce_{0.6}Fe_{0.4}O_{2-\delta}$ sample, confirming once again the high activity of the Fe-doped catalyst.

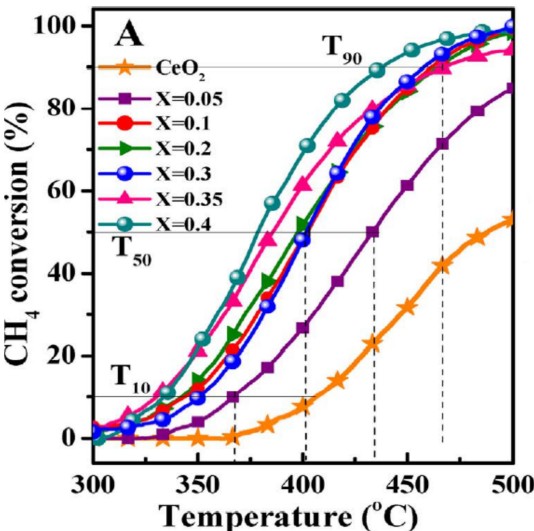

**Figure 19.** Methane combustion over Fe-doped ceria catalysts at different temperatures. Reprint from [165]. Copyright (2018), with permission from Elsevier.

The two most active Fe-loaded samples (x = 0.3 and 0.4) were tested for catalytic durability at a conversion of 75% for 50 h on stream. During the combustion, $Ce_{0.7}Fe_{0.3}O_{2-\delta}$ showed high stability with only a slight decrease in activity from 75% to around 70%; however, for the $Ce_{0.6}Fe_{0.4}O_{2-\delta}$ sample a significant drop in methane conversion (from 75% to 50%) was observed. This difference in behavior is ascribed to the growth of additional surface $Fe_2O_3$ particles in $Ce_{0.6}Fe_{0.4}O_{2-\delta}$ compared to the $Ce_{0.7}Fe_{0.3}O_{2-\delta}$ sample, as evidenced by Raman analysis. Considering these results, the iron oxide particles are highly active, but less stable on stream, deactivating the catalyst, whereas the $Ce_{0.7}Fe_{0.3}O_{2-\delta}$ sample with only Ce–Fe–O solid solution retains most of its activity under continuous operation, being more stable [167].

This work presented a series of Fe-doped ceria catalysts with excellent catalytic activity, with very high methane conversion below 450 °C, indicating the synergetic play between the free $Fe_2O_3$ nanoparticles and the oxygen vacancies as more iron is loaded. In addition, these results suggest that the catalytic performance in methane combustion is more durable in the case of catalysts based on mixed oxide solid solutions with oxygen vacancies, without other free oxide particles on the surface.

Specific studies on monometallic doped ceria materials have focused on manganese as doping element for its excellent redox properties in oxidation reaction which were prepared through two different methods: redox precipitation (RP) and co-precipitation (CP) [168]. The redox precipitation is an interesting approach where the mixed oxide materials, MnCe-RP, are obtained via redox titration of $Mn(NO_3)_2$ with a solution of $(NH_4)_2Ce(NO_3)_6$ and $KMnO_4$ in basic conditions, after which they are calcined in air at 500 °C for 6 h. The co-precipitation method afforded the mixed oxides noted MnCe-CP, from precipitates of $Mn^{2+}$ and $Ce^{4+}$ nitrates with KOH solution, calcined with the previous thermal program.

These methods have produced mixed oxide materials with different textural characteristics as is evidenced by BET surface area and pore volume of 111 $m^2/g$ and 62.38 $cm^3/g$, respectively, for MnCe-RP, larger than those of MnCe-CP (47 $m^2/g$ and 31.74 $cm^3/g$). The composition of Mn-Ce samples was evaluated by EDS and XRF analysis which indicated high amounts of Mn in both materials with atomic ratios of Mn/Ce for MnCe-RP (11.64) and MnCe-CP (8.42), and additionally large content of K (11.5%) in the sample prepared by RP. More differences appear in XRD patterns where peaks of $CeO_2$ and $Mn_5O_8$ are observed for MnCe-CP, while the presence of $K_xMn_8O_{16}$ is highlighted in MnCe-RP which would explain the high K content.

The catalytic activity of the Mn-Ce samples was tested through methane oxidation reaction with a feed gas consisting of 1% $CH_4$ and 10% $O_2$, balanced with $N_2$. The tests indicated that the redox precipitation afforded a more active catalyst than the co-precipitation method as MnCe-RP exhibited a $T_{50}$ of 446 °C, whereas MnCe-CP had a $T_{50}$ of 469 °C, since the initial difference of specific surface area provides more active sites in the former catalyst. XPS analysis also revealed that the amount of $Mn^{4+}$, $Ce^{3+}$ and surface lattice oxygen is higher in the redox precipitated sample compared to the co-precipitated one, which greatly enhances the catalytic performance in methane combustion [169,170]. Furthermore, the K ion from the $K_xMn_8O_{16}$ phase could improve the reducibility of the MnCe-RP catalyst as K ions may weaken Mn–O bonds to facilitate the release of oxygen from the lattice required in the methane oxidation reaction.

The stability of MnCe-RP and MnCe-CP catalysts in lean methane combustion at 600 °C for more than 21 h on stream was evaluated to show the excellent performance in time. The first catalyst preserved its initial methane conversion, dropping from 100% to 99.94% after 21 h, whereas MnCe-CP had a more pronounced decrease in activity as the conversion lowered until 92.8% after the combustion process, thus proving the superiority of the redox precipitated MnCe material in terms of catalytic durability.

Another approach to improve the performance of ceria materials in lean methane combustion consists of the preparation of $NiO/CeO_2$ catalyst via impregnation of ceria support with nickel nitrate to achieve different NiO loadings from 1 to 20 wt% [171]. XRD patterns of NiO-supported materials revealed the presence of $CeO_2$ phase, and at loading of 7 wt% or greater the presence of NiO cubic phase, which was highly dispersed on the ceria surface as the peaks for NiO were still weak even at 20 wt% NiO loading. Moreover, no shifts or evident broadening are observed in $CeO_2$ phase peaks, indicating that most of the $Ni^{2+}$ ions aggregated in particles on the support surface, without considerable incorporation into the $CeO_2$ lattice, thus the impregnation method is not a technique to favor the insertion of other metal ions in the ceria structure. The crystallites did not change much in size after the impregnation, being around 8–9 nm for all the samples. However, the surface area decreased as the Ni content became higher, from 76.4 $m^2/g$ for pure ceria to 55.5 $m^2/g$ for 20 wt% $NiO/CeO_2$.

Catalytic combustion of gas mixture with 1% $CH_4$ and 4% $O_2$ in Ar over the NiO doped materials showed a great enhancement of activity in methane conversion compared to pure $CeO_2$ with high $T_{10}$, $T_{50}$ and $T_{90}$ of 456, 540 and 615 °C, respectively, whereas the most active catalyst, the 10 wt% $NiO/CeO_2$, had low $T_{10}$, $T_{50}$ and $T_{90}$ of 333, 415 and 467 °C, respectively (Figure 20).

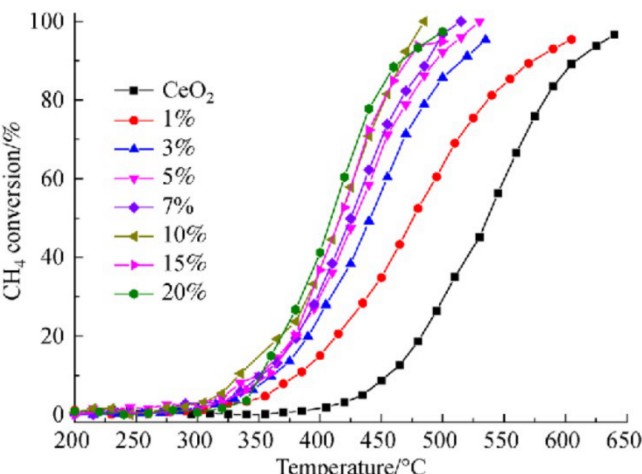

**Figure 20.** Catalytic activity of ceria-supported NiO materials in methane combustion. Reprint from [171]. Copyright (2020), with permission from Springer Nature.

The enhanced catalytic activity is attributed to the increase of NiO reactive sites due to the growth of more NiO particles over ceria supports; however, additional improvement in activity is not observed when the NiO loading exceeds 10%, as the NiO particles start to agglomerate on the surface with no increase in the number of active sites. Furthermore, the long term stability of the 10 wt% $NiO/CeO_2$ catalyst was also investigated by on stream tests at 600 °C for 150 h which exhibited a stable conversion of 100% without any decrease, suggesting its excellent stability in $CH_4$ combustion.

After the stability test, the catalyst showed no changes in the XRD patterns of NiO and $CeO_2$ phases, indicating that the NiO particles retain their high dispersion on the support. This behavior proves a synergetic effect between NiO and $CeO_2$ whose sintering at high temperature was prevented by favorable interactions between the oxides which are usually known to be sintered easily at high temperature, resulting in deactivation during methane oxidation [172]. The minimal insertion of $Ni^{2+}$ ions into the $CeO_2$ lattice which increases the activity and oxygen vacancy concentration of the catalyst, in addition to the high dispersion of NiO particles on the support surface, led to the design of a highly stable and active catalyst.

Several studies regarding the stability of $Co_3O_4$ at high temperature were directed towards the preparation of $Co_3O_4/CeO_2$ composite oxides containing 30 wt% of $Co_3O_4$ by a co-precipitation method, followed by calcination at 650 C for 5 h [173]. This method allowed a good dispersion of $Co_3O_4$ particles in the mixed oxide, besides achieving smaller grain size, as the prepared material had a surface area of 36 $m^2/g$ and crystallite size of 15 nm, compared to precipitated pure $Co_3O_4$ with a surface area of 9 $m^2/g$ and crystallite size of 89 nm. After an ageing treatment at 750 °C for 7 h, both samples showed signs of sintering in a considerable manner, especially for the pure $Co_3O_4$, which halved its surface area to 4.7 $m^2/g$ and doubled its crystallite size to 155 nm, whereas the $Co_3O_4/CeO_2$ material preserved to a great extent its textural features with surface area of 30 $m^2/g$ and grain size of 27 nm.

Further methane combustion tests revealed that pure $Co_3O_4$ and $Co_3O_4/CeO_2$ had high activity as $T_{50}$ values were 387 and 380 °C, respectively. However, the aged samples exhibited lower activities than the fresh counterparts since the $T_{50}$ values were 447 and 395 °C. From the changes in textural properties and the $T_{50}$ values, a stabilization effect can

be observed of the ceria support over $Co_3O_4$, as the increased surface area of the composite oxides allows a high dispersion of the crystalline $Co_3O_4$ phase. At the same time, this work showed the impact of high temperature in the range of 750–800 °C over the pure $Co_3O_4$ in fresh and aged form where the methane conversion started to drop notably, whereas the $Co_3O_4$ /$CeO_2$ catalyst preserved most of its activity even in the aged form, confirming the role of the ceria support in stabilizing the active $Co_3O_4$ phase.

Exploring the effect of the synthesis method of other Co-doped ceria systems on the catalytic performance in complete methane oxidation led to the preparation of two series of catalysts with Co content ranging from 2 to 15 wt% via an incipient wetness impregnation method on ceria supports manufactured by hydrothermal and precipitation technique [174]. The mixed oxides with supports produced by the hydrothermal method showed superior textural characteristics with BET surface area between 80–100 $m^2$/g and $CeO_2$ particle size of around 6–9 nm, compared to those obtained through the precipitation method which have surface areas of 39–43 $m^2$/g and ceria particle size of 16–20 nm. Taking into account these results, a deposition of $Co_3O_4$ particles is suggested over ceria surface, without extensive insertion of Co ions into the $CeO_2$ lattice. Moreover, for the more Co loaded materials (15%), the crystallite size of $Co_3O_4$ is smaller on hydrothermal ceria (16 nm) than on precipitated ceria (76 nm), which accordingly impacts the catalytic activity.

As expected, incorporation of Co on hydrothermal prepared ceria supports afforded more active catalysts in lean combustion of feed gas with 0.5% $CH_4$, 10% $O_2$, balanced by He, than over precipitated ceria. 15 wt% Co/$CeO_2$-H sample was the most active from the composite metal oxides and was able to reach methane conversion of 50% and ~90% at 520 and 600 °C, respectively, in line with TPR measurements, although less active than pure $Co_3O_4$ material. Nevertheless, catalytic stability tests over time at 600 °C on the 15 wt% Co/$CeO_2$-H sample showed a relatively stable conversion of methane after 10 h, which decreases from 88% to 81%, indicating notable tolerance to thermal ageing and sintering. The thermal resistance of the catalyst is attributed to the intimate contact between the Co and ceria species, thanks to the high specific surface area.

More impressive is the stability of 15 wt% Co/$CeO_2$-H over methane catalytic conversion in the presence of water vapor which is generally known to deactivate combustion catalysts, representing a challenge to be overcame for further applications. The Co-doped ceria material displayed no considerable inhibition effect when water vapor was added in the gas mixture, as its catalytic performance slightly decreases. Indeed, the $T_{50}$ values only increase by less than 20 °C when the water content reached 10% (Figure 21).

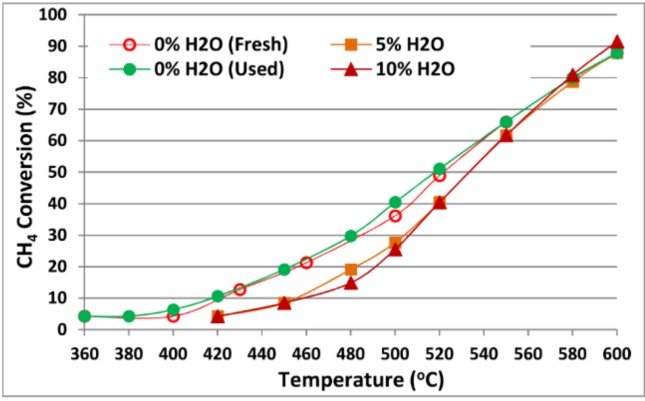

**Figure 21.** $CH_4$ catalytic combustion over 15 wt% Co/$CeO_2$-H at different water vapor content in gas mixture (0%, 5% and 10%). Reprinted from [174].

These results advocate the synergetic effects of ceria supports and $Co_3O_4$ nanoparticles which afforded a promising composite material for complete methane oxidation, especially as it displayed excellent water resistance properties in $CH_4$ combustion in wet conditions.

Among the aforementioned monometallic doped ceria systems, the most attractive catalysts with high performance and relative stable methane combustion are those containing Cu ($T_{50} \cong 350$ °C), Fe ($T_{50} = 378$ °C), Ni ($T_{50} = 415$ °C) and Co ($T_{50} = 380$ °C).

### 4.2. Bimetallic Modified Ceria Catalysts

Extensive attempts to improve the activity of ceria materials in lean methane combustion have focused on the integration of multiple heteroatoms in $CeO_2$ structure in order to improve the textural and redox properties which have a crucial role in catalytic performance.

A study centered on Zr-doped ceria materials used as supports for Ni in dry impregnation showed the superiority in complete methane combustion of Ni containing catalysts compared to the high Zr doped ceria materials which oxidize large amounts of hydrocarbon to CO. Considering that $Ce_{0.83}Zr_{0.17}O_2$ support preserves its surface area after the calcination process at 600 C (from 88 to 61 m$^2$/g), the impregnation with Ni afforded the highly active catalyst 2 wt% Ni/$Ce_{0.83}Zr_{0.17}O_2$ with $T_{10}$, $T_{50}$ and $T_{90}$ values of ~375, 452 and 532 °C, respectively, performing closely to the 2 wt% Ni/$CeO_2$ (375, 438 and 523 °C, respectively) [175]. The variation of apparent activation energy ascertained that the deposition of Ni on Zr-doped ceria supports notably decrease the energetic barrier for methane combustion as can be evidenced from the undoped Ni materials with $E_a$ around 120 kJ/mol and the Ni-doped catalyst with $E_a$ around 90 kJ/mol. Moreover, the calculation of turnover frequency (TOF) revealed that 2 wt% Ni/$Ce_{0.83}Zr_{0.17}O_2$ had a TOF of 0.45 s$^{-1}$ which is greater than the TOF of Ni/$CeO_2$ (0.245 s$^{-1}$), although they displayed similar activities in terms of conversion.

Taking into account this result, the difference resides in the specific surface area which is larger for the $Ce_{0.83}Zr_{0.17}O_2$ support than pure ceria, thus providing more active sites of Ni for $CH_4$ activation. As the combustion process follows the Mars van Krevelen mechanism, it can be argued that methane molecules reduce NiO species to $Ni^0$ which are further re-oxidized by the lattice oxygen from the support. Therefore, the doping of ceria with Zr which improves the reducibility of the support, together with the high activity of deposed NiO particles in hydrocarbon activation, led to the formation of an active catalyst with high performance in methane combustion.

Another catalyst consisting of deposited Ni onto Zr-doped ceria support through a wet impregnation method was tested at the temperature range of 300–800 °C for methane combustion with a feed gas of 3% $CH_4$, 10% $O_2$ and He in balance. Exhibiting an average BET surface area of 77.9 m$^2$/g and no observable separate phases of Ni and NiO in XRD patterns, the 5 wt% Ni/$Ce_{0.75}Zr_{0.25}O_2$ catalyst showed a significant increase of activity with a $T_{50}$ value of 480 °C, compared to the $Ce_{0.75}Zr_{0.25}O_2$ support with $T_{50}$ of 590 °C (Figure 22) [172].

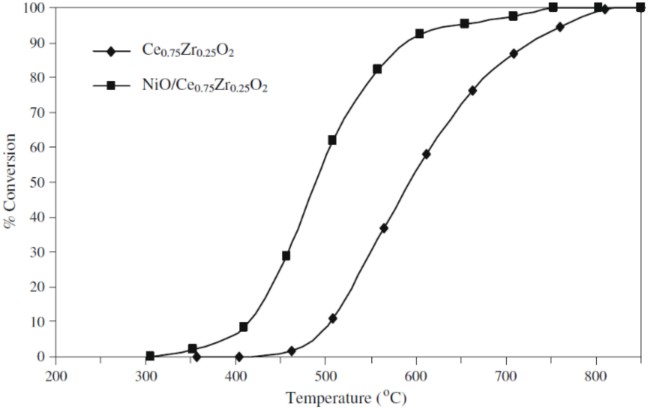

**Figure 22.** Methane conversion over $Ce_{0.75}Zr_{0.25}O_2$ and NiO/$Ce_{0.75}Zr_{0.25}O_2$ catalysts. Reprint from [172]. Copyright (2009), with permission from Elsevier.

All these results indicated the potential of Ni modifying Ce-Zr systems to enhance considerably the catalytic performance in methane combustion by providing more reactive sites required for $CH_4$ activation, as long as the modification process does not diminish to a great extent the specific surface area of the final materials.

Doping of Ce-Zr mixed oxides with Sc in atomic content of 2%, 4% and 6%, respectively, has proved to be quite unsuccessful in obtaining a performing catalyst in methane combustion, even though the preparation through sol-gel citrate route afforded materials with good textural features (BET surface areas of 35–48 $m^2/g$ and crystallite size of around 10 nm) [176]. Although, the addition of Sc ions in the Ce-Zr lattice improved slightly the catalytic activity of Zr-doped ceria by increasing reducibility, unfortunately the Sc containing samples displayed still high light-off temperatures of ~550 °C and $T_{50}$ values of ~700 °C, as Sc is not active redox to bring more activity to mixed oxide systems.

An interesting approach to obtaining bimetallic ceria systems involves the synthesis of meso-structured $Co_3O_4$ by thermal treatment of $Co(NO_3)_2 \cdot 6H_2O$ using as template KIT-6, a mesoporous silica with large, uniform and accessible pores, and a further wet impregnation of the Co material with cerium nitrate dissolved together with La or Mn nitrate in ethanol [177]. In this way, 10% at. La- or Mn-doped ceria particles infiltrate the pores of the meso-$Co_3O_4$ to afford materials with specific surface area of 67 and 55 $m^2/g$, respectively, and average crystallite size for $Co_3O_4$ and $CeO_2$ of around 130 and 73 nm, respectively.

Considering that the meso-$Co_3O_4$ displayed high activity in lean methane combustion with $T_{50}$ of 390 °C, the composite materials containing doped ceria also showed similar activity, however with larger $T_{50}$ values of 400 °C for meso-$Co_3O_4$/10%La-$CeO_2$ and 445 °C for meso-$Co_3O_4$/10%Mn-$CeO_2$. Nevertheless, the combustion tests are performed with lean gas mixture of 0.1% $CH_4$ at high weight hourly space velocity (WHSV) of 180,000 mL/g/h, significantly higher than the most usual space velocities used in catalytic tests over similar catalysts, which can evidence that the as-prepared materials are more active for methane oxidation. This association proved to be unfortunate as the addition of ceria particles reduces the specific surface area of the material, decreasing the activity slightly. However, most of the catalytic activity remains compared to meso-$Co_3O_4$, offering alternatives to the list of support materials for noble metals or other active elements.

A study focusing on catalysts with 12 wt% Ni deposited through wet impregnation on composite support consisting of $CeO_2$ and MgO in different ratios produced via citrate sol-gel method revealed that the most active system in methane combustion (5% $CH_4$ in air) after two evaluation tests is the Ni/$(CeO_2)_{0.1}$-$(MgO)_{0.9}$ catalyst with the highest surface area (64 $m^2/g$) and smallest crystallite size of 19 nm from the series (Figure 23) [178].

Initially, Ni/$CeO_2$ was the most active in the first run ($T_{10}$ = 347 °C, $T_{50}$ = 432 °C, $T_{90}$ = 512 °C); however, after the second run, the activity of Ni/$CeO_2$ decreases, reaching $T_{10}$, $T_{50}$, $T_{90}$ of 380, 469 and 544 °C, respectively, whereas Ni/$(CeO_2)_{0.1}$-$(MgO)_{0.9}$ catalyst obtained lower temperatures in average ($T_{10}$ = 384 °C, $T_{50}$ = 458 °C, $T_{90}$ = 528 °C), suggesting an increase in thermal stability for the catalyst containing MgO. XRD patterns did not present separated NiO phases in the sample except NiO/$CeO_2$, indicating that NiO and MgO can easily form a solid solution as NiO is highly dispersed on the supports and both of them have similar cubic structures and lattice parameters [179]. The formation of the Ni-Mg-O solid solution ascertained the better resistance to sintering and the increase in thermal stability of the catalyst.

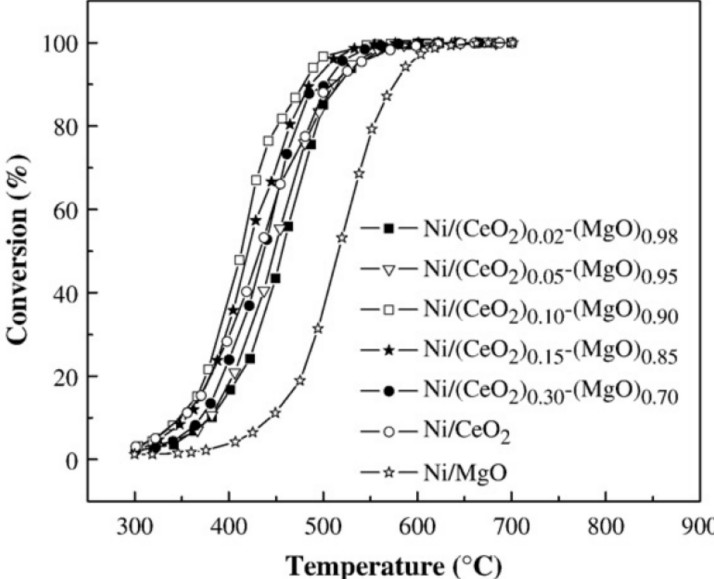

**Figure 23.** Conversion of methane over Ni/$(CeO_2)_x$-$(MgO)_{1-x}$ catalysts (first evaluation). Reprint from [178]. Copyright (2009), with permission from Elsevier.

After it was determined as the best composition for the support, the activity of other transition metal catalysts (Ni, Co and Cu) with the same loading of 12 wt% was compared, with interesting results (Figure 24). Despite having similar activities in the first evaluation, the second evaluation revealed that the catalyst with nickel displayed the best thermal stability ($\Delta T_{50} = 45$ °C), whereas Cu/$(CeO_2)_{0.1}$-$(MgO)_{0.9}$ showed the most drastic drop in activity ($\Delta T_{50} = 120$ °C) and Co/$(CeO_2)_{0.1}$-$(MgO)_{0.9}$ with an intermediate stability ($\Delta T_{50} = 72$ °C) (Figure 24).

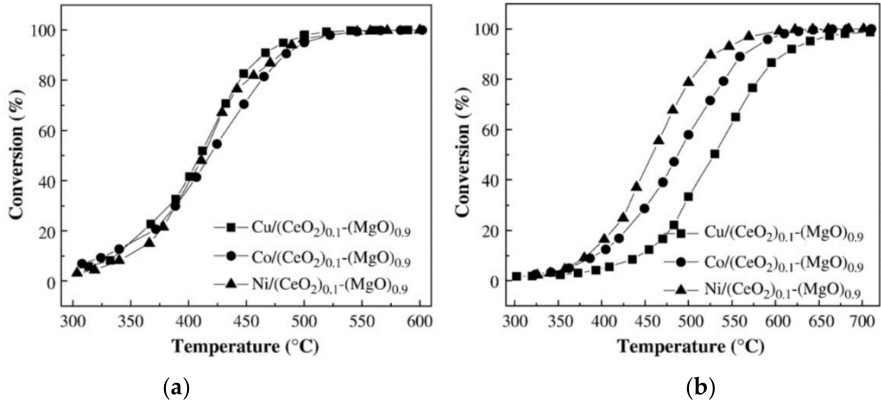

**Figure 24.** Conversion of methane over M(Ni, Co, Cu)/$(CeO_2)_{0.1}$-$(MgO)_{0.9}$ catalysts in first (**a**) and second run (**b**). Reprint from [178]. Copyright (2009), with permission from Elsevier.

Taking into account this significant decrease, it can be argued that Cu/$(CeO_2)_{0.1}$-$(MgO)_{0.9}$ suffered a sintering process where CuO agglomerated on the catalyst surface, as inferred by the severe decrease of BET surface area (from 64.6 to 6.6 m$^2$/g). In the case of Ni and Co catalysts, the surface areas were still similar after the second evaluation (30.9 and 33.6 m$^2$/g, respectively). This work showed the preparation of a highly active catalyst by the successful association of $CeO_2$ and MgO which confers good thermal stability to the support for Ni, a transition metal endowed with high activity for methane combustion.

A series of novel catalysts has been synthesized via four different methods: impregnation (IWI), microwave in-situ grown (MW), impregnation-combustion (CS) and plasma treatment (P) to deposit Co-Al-O composite on ceria support prepared by solu-

tion combustion method. The catalysts, denoted as $CoAlO_x/CeO_2$-X (X stands for the preparation method), achieved a 30 wt% loading of Co-Al-O composite in the material with a Co/Al ratio of 3:1 [31]. XRD analysis revealed a higher dispersion of $Co_3O_4$ as the diffraction peaks were broader and weaker in $CoAlO_x/CeO_2$-X (X = MW, CS, P) than those in $CoAlO_x/CeO_2$-IWI catalyst. At the same time, the IWI catalyst had the smallest BET surface area (20.1 $m^2$/g) and largest $Co_3O_4$ crystallite size (26.8 nm), compared to the other catalysts which presented similar textural properties.

Further, their catalytic performance in methane combustion was evaluated in the temperature range of 300–670 °C using a gas stream of 10% $CH_4$ and 25% $O_2$ in Ar (Figure 25). $CoAlO_x/CeO_2$-P showed the highest activity in methane oxidation among the series with $T_{10}$, $T_{50}$, $T_{90}$ of 335, 415 and 580 °C, respectively, whereas the IWI catalyst had the higher conversion temperatures ($T_{10}$ = 460 °C, $T_{50}$ = 511 °C, $T_{90}$ = 614 °C), as expected from the available lower surface area and poorer dispersion of Co species. Moreover, the preparation technique impacted the apparent activation energy, which was reduced considerably from 143 kJ/mol for pure ceria to 92 kJ/mol for $CoAlO_x/CeO_2$-P.

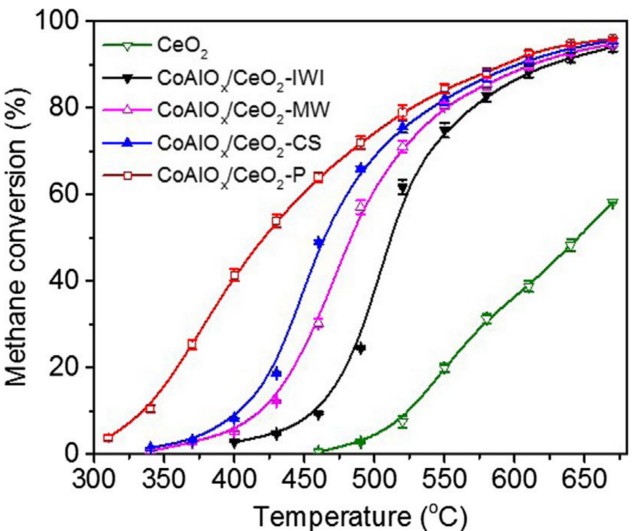

**Figure 25.** Methane conversion over $CeO_2$ and $CoAlO_x/CeO_2$ versus temperature. Reprint from [180]. Copyright (2018), with permission from Elsevier.

The superiority of the plasma treatment as technique for catalyst preparation was ascertained with the help of XPS analysis which indicated for the $CoAlO_x/CeO_2$-P catalyst the highest $Co^{3+}/Co^{2+}$ (0.647), $Ce^{3+}/Ce^{4+}$ (0.353), $O_{ads}/O_{latt}$ (1.114) and Co/Ce (8.13) surface ratios, explaining the improved catalytic activity by better reducibility and oxygen mobility, higher concentration of active oxygen species and higher dispersion of Co species on the surface.

In addition, the stability of $CoAlO_x/CeO_2$-P and $CoAlO_x/CeO_2$-IWI catalysts was assessed for 3000 min of oxidation reaction at 580 °C (Figure 26). Considering the results of the stability test, it can be argued that $CoAlO_x/CeO_2$-P displayed a more stable catalytic activity as it decreased from 90% to 78%, whereas the $CoAlO_x/CeO_2$-IWI had a significant decrease of activity from 71% to 42%, thus proving the synergetic effect arising from the interaction between the support and the Co species, associated with the use of plasma treatment.

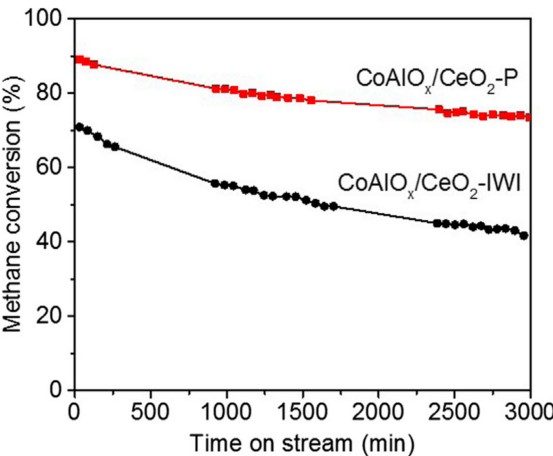

**Figure 26.** Methane combustion with time-on stream over $CoAlO_x/CeO_2$-X (X = P, IWI) at 580 °C. Reprint from [180]. Copyright (2018), with permission from Elsevier.

Other works studied cobalt catalysts with a Co loading of 30 wt% deposited on mixed supports of $Al_2O_3$ and $CeO_2$ with different cerium loadings of 5, 10, 15, 20 and 30 wt%, prepared by a precipitation method with $Na_2CO_3$ [181]. The prepared catalysts presented BET surface areas between 73 and 102 $m^2/g$; however, a loss in surface area is observed, compared to Co/bare alumina (108 $m^2/g$), with the increase in cerium loading. Nevertheless, the crystallite size of $CeO_2$ and spinel-like cobalt phase ($Co_3O_4$ and/or $CoAl_2O_4$) was similar in all the samples, at around 8–14 nm and 23–24 nm, respectively, as evidenced by the wide peaks visible in XRD patterns, suggesting a good dispersion of ceria and cobalt spinel particles on the alumina support.

Besides the specific bands of $Co_3O_4$, Raman spectra on the cobalt catalysts revealed two signals (705 and 725 $cm^{-1}$) ascribed to the presence of cobalt aluminate whose intensity decreased when ceria was deposited in higher amounts [182]. Taking into account this result, it seems that deposited ceria in the material acted as a barrier between Co and Al species hindering the formation of $CoAl_2O_4$ which is undesirable for its poor reducibility, favoring the maintenance of a highly active $Co_3O_4$ phase.

The activity in methane combustion of the Co-doped ceria-alumina was examined in the 200–600 °C temperature range with feed gas of 1% $CH_4$ and 10% $O_2$, balanced by $N_2$, indicating higher conversions than 95% at 600 °C for all the Co/xCe-Al tested catalysts (Figure 27). Light-off curves identified the most active catalyst to be Co/20Ce-Al with a $T_{50}$ of 480 °C, as inferred from the XPS analysis of the material surface which revealed the highest content of surface Co (32.9%) with the highest Co/(Ce + Al) and $Co^{3+}/Co^{2+}$ molar ratios, responsible for the active sites in combustion.

The incorporation of cerium ions in the material prevented the formation of cobalt aluminate, as the $Co/Al_2O_3$ had the lowest Co/(Ce + Al) surface ratio (0.32), indicating a diffusion of Co ions into the alumina lattice, whereas the Co/20Ce-Al with a Co/(Ce + Al) surface ratio of 0.84 presented more surface cobalt with less $CoAl_2O_4$ being formed.

Moreover, the increase of $Co^{3+}/Co^{2+}$ molar ratio when more cerium is loaded suggests an insertion of $Ce^{4+}$ cations into the $Co_3O_4$ lattice as more $Co^{2+}$ ions migrate into the bulk to maintain the charge balance and $Co^{3+}$ concentrate on the oxide surface. All these results indicate that an optimum was achieved for the Co/20Ce-Al catalyst.

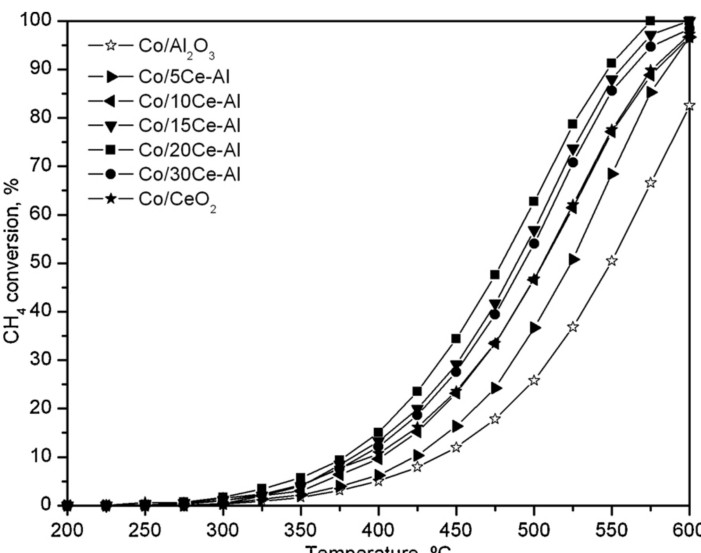

**Figure 27.** Light-off curves of the Co/xCe-Al catalysts. Reprint from [181]. Copyright (2020), with permission from Elsevier.

Notably, the Co/20Ce-Al catalyst exhibited a good thermal stability in dry conditions for an extended reaction time of 60 h at 450 °C, followed by 90 h at 525 °C, since a relatively constant methane conversion of around 30% was observed at 450 °C with no signs of deactivation. However, at increased temperature of 525 °C, the catalyst started to lose activity as the conversion dropped from 78% to 64% in 25 h, and to 58% in the next 25 h, then remained stable for the last 40 h during the stability test.

Consequently, $CeO_2$ had a stabilizing effect on the $Co_3O_4$ phase and also improved the reducibility and oxygen mobility of the support to afford a highly active catalyst for methane combustion.

The results in methane oxidation obtained from bimetallic doped ceria catalysts are promising, achieving more thermal stability for a prolonged operating time, as well as increased selectivity towards the complete combustion of methane.

### 4.3. Multimetallic Modified Ceria Catalysts

An interesting approach to combine multiple cations besides ceria into some materials has sought to achieve competitive and active catalysts for complete methane combustion as alternative for noble metal catalysts. The synthesis of mixed oxides derived from LDH via the co-precipitation method and calcination at 750 °C in air afforded a series of Cu-Ce-MgAlO catalysts with fixed 10 at.% Ce, Mg/Al atomic ratio of 3 and different Cu content ranging from 6% to 18 at.%, with respect to cations [183].

The textural properties characterization of the Cu(x)CeMgAlO catalyst showed a decrease in surface area from 169 $m^2/g$ for Cu(6) sample to 108 $m^2/g$ for Cu(18) sample, still high thanks to the MgAlO mixed oxide as support, and an increase in $CeO_2$ crystallite size with the increase of Cu loading from 6.3 nm for Cu(6) sample to 15.1 nm for Cu(18). Taking into consideration that the presence of CuO phase was only observed in the Cu(18)CeMgAlO material in XRD patterns, it can be concluded that in the samples with Cu content lower than 18 at.%, Cu-doped ceria particles are highly dispersed in the Mg(Al)O matrix, whereas at high Cu loading (18 at.%), larger $CeO_2$ crystallite coexist with CuO particles as a separate phase identified by XRD.

After combustion tests with gas stream of 1% $CH_4$ in air, the most active catalyst in the series turned out to be Cu(15)CeMgAlO mixed oxide with $T_{10}$, $T_{50}$ and $T_{90}$ of 380, 463 and 540 °C, respectively, having a $T_{50}$ higher at ca. 45 °C than that corresponding to an industrial $Pd/Al_2O_3$ catalyst, displaying excellent catalytic performance (Figure 28). Moreover, in matters of total methane conversion, the Cu(15) and Cu(18) systems have shown $T_{100}$ at around 600 °C, temperature only 30 °C higher than that of the $Pd/Al_2O_3$.

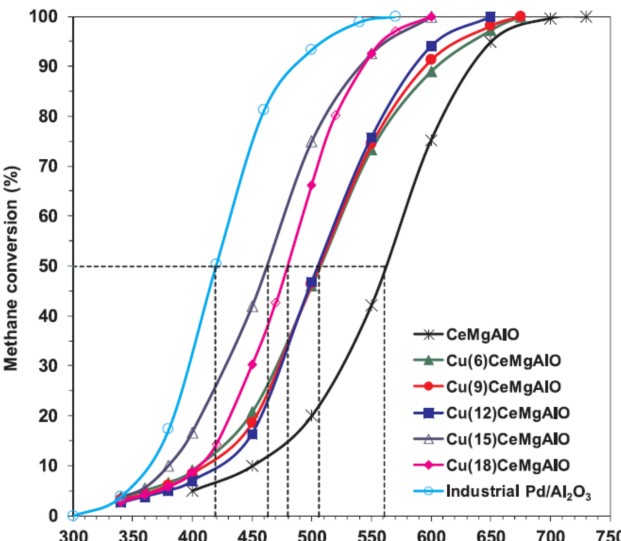

**Figure 28.** The light-off curves for the combustion of methane over Cu(x)CeMgAlO catalysts. Reprint from [183]. Copyright (2019), with permission from Elsevier.

XPS data confirmed the high activity of Cu(15)CeMgAlO mixed oxide determined through catalytic combustion tests in the light-off curves, indicating the highest surface $Cu^{2+}$/Cu atomic ratio of 0.9 in the series, with a consequence on the number of active sites represented by $Cu^{2+}$ ions, along with the high dispersion of copper in the catalyst.

The catalytic stability tests showed that Cu(15)CeMgAlO material, the most active in the series studied, is capable of maintaining methane conversion at around 92% for more than 50 h on stream at 550 °C with no evidence of decrease in activity, proving once again the stabilizing effect of the Mg(Al)O support over the active Cu-doped ceria particles (Figure 29).

Taking into account these results, it can be argued that there is a synergetic interaction between copper and cerium species, together with mixed oxide support for thermal stability, which requires the identification of an optimum to achieve an enhancement of the catalytic activity in methane combustion.

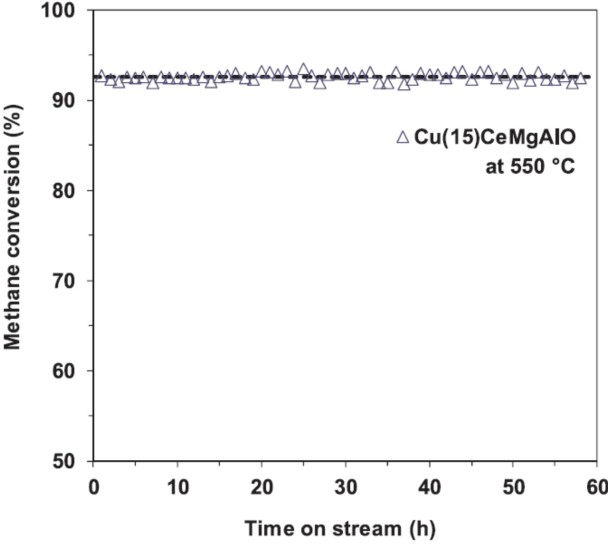

**Figure 29.** Conversion of methane versus time on stream over Cu(15)CeMgAlO catalyst. Reprint from [183]. Copyright (2019), with permission from Elsevier.

Further studies were realized on CuCeMgAlO systems to improve its catalytic performance in methane combustion to get even closer to that of the industrial $Pd/Al_2O_3$.

Through LDH precursor synthesized by co-precipitation, a series of catalysts comprised of four M(3)CuCeMgAlO mixed oxides containing 3 at.% M (M = Mn, Fe, Co, Ni), 15 at.% Cu, 10 at.% Ce (at.% with respect to cations), with Mg/Al atomic ratio of 3 has been obtained [184].

Doping the CuCeMgAlO with an additional transition metal afforded the enhancement of the catalytic activity for the initial system in all cases by reducing the conversion temperatures by several degrees, as evidenced from the light-off curves depicted in Figure 30. Among the series, the Co(3)CuCeMgAlO catalyst showed itself to be the most active one with $T_{10}$, $T_{50}$ and $T_{90}$ of 372, 438 and 511 °C, respectively, its $T_{50}$ being lower with 25 °C than that of the M-free CuCeMgAlO mixed oxide ($T_{50}$ = 463 °C) and only ca. 20 °C higher than that of the reference Pd/Al$_2$O$_3$ catalyst with $T_{50}$ and $T_{90}$ values of 419 and 484 °C, respectively.

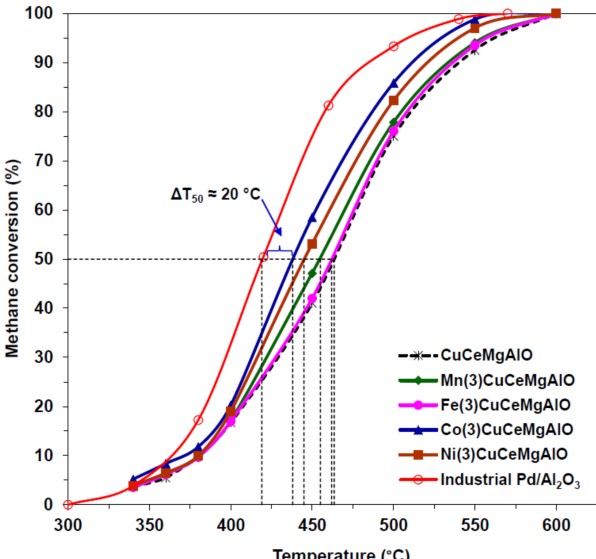

**Figure 30.** The light-off curves for methane conversion over M(3)CuCeMgAlO catalysts with feed gas of 1% CH$_4$ in air. Reprinted from [184].

Considering that the catalyst reducibility has increased thanks to further doping with all the tested metals, it can be argued that the insertion of Co$^{3+}$ ions into the material proved to be critical in relation to the catalytic performance, suggesting a key role of the synergetic interactions between Co$^{3+}$, Cu$^{2+}$ and Ce$^{4+}$ species.

As its free dopant counterpart, the Co(3)CuCeMgAlO catalyst displayed good stability during the total oxidation of methane at 520 °C for 60 h with no observable change in the catalytic performance due to its good thermal resistance provided by Mg(Al)O matrix.

A very interesting study focusing on the reaction mechanism has been reported on CeO$_2$-ZrO$_2$ mixed oxides [185]. The model corresponds to the L-H mechanism with co-adsorption of O$_2$ and CH$_4$ on the same sites formed by a metal-oxygen pair on the oxide surface [154]. This mechanism leads to the best fitting for the kinetics of CH$_4$ oxidation over CeO$_2$-ZrO$_2$ mixed oxides. The resulting rate expression is analogous to that of the equation with terms of O$_2$ adsorption in P$_O^{1/2}$ instead of P$_O$ [154]:

$$r_{\text{LH}} = k_C \frac{K_C P_C \cdot K_O P_O}{(1 + K_C P_C + K_O P_O)^2} \tag{13}$$

It was found that oxygen was more strongly adsorbed than methane, adsorption energy of oxygen on Ce$_{0.75}$Zr$_{0.25}$O$_2$ being 2.4 times higher than that of methane in the 400 °C–600 °C temperature range of reaction. The examination of these results [154,185]

shows that the L-H model was ascertained on the basis of the coefficient of correlation ($R^2$ = 0.994) better than that of an E-R model with $CH_4$ reacting on adsorbed oxygen [154]:

$$r_{ER} = k_O \frac{K_O P_O \cdot P_C}{(1 + K_O P_O)} \quad (14)$$

$R^2$ = 0.929 for this equation with a term in $P_O^{1/2}$ instead of $P_O$.

However, Belessi et al. [186] showed that the $CH_4$ oxidation reaction on ferrite-like perovskite catalysts obeyed an E-R mechanism, rather than a L-H model [186]. They also demonstrated that the E-R rate equation could be assimilated to power-law kinetics with apparent orders close to the experimental values [154]:

$$r_{ER} = k_O \frac{(K_O P_O)^{1/2} \cdot P_{CH_4}}{1 + (K_O P_O)^{1/2}} \approx k'_O (K_O P_O)^{0.5-a} P_{CH_4}^b \quad (15)$$

Experimental orders, measured in the 420 °C–620 °C temperature range, are between 0 and 0.5 for $O_2$ and close to 1 for methane.

Methane oxidation requires high temperature of reaction, above 400 °C for most oxides. Oxygen mobility can thus be very high, which allows MvK mechanisms to occur. For instance, over $Co_3O_4$, Zavada et al. [154,187] showed that $CH_4$ oxidation shifted gradually from a supra-facial mechanism of the E-R type at 400 °C to a MvK mechanism implying cobalt-oxo species above 600 °C [187]. Changes of mechanism may also occur depending on the catalyst composition. $CH_4$ oxidation was studied over a series of $Ce_xSn_{(1-x)}O_2$ catalysts (Figure 31) [188] and the authors concluded that the MvK model was preferred over the Sn-rich catalysts [154,188].

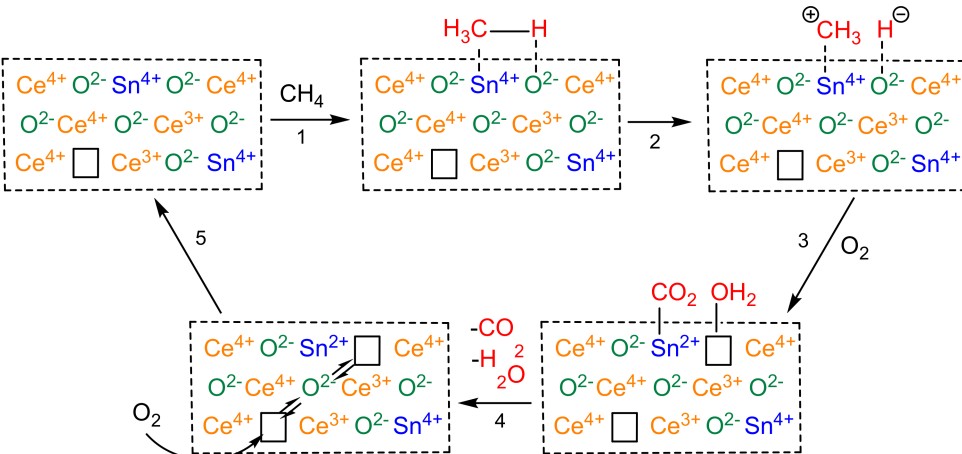

**Figure 31.** Mars and van Krevelen (MvK) mechanism of methane oxidation over $Ce_xSn_{(1-x)}O_2$ catalysts. Adapted from [188].

It must be noted that methane is the most difficult hydrocarbon molecule to oxidize due to its high stability [75], being often used as a test molecule in catalytic combustion processes. Although the mechanism of its combustion on the catalyst surface is not well established, it is accepted that the activation step proceeds through the hemolytic abstraction of a hydrogen atom from the hydrocarbon (dissociative chemisorption of the hydrocarbon), followed by a fast sequence of reactions between the alkyl group and oxygen species coordinated on the solid surface [189,190]. Two different mechanisms were hypothesized for methane catalytic combustion: one at low temperatures involving chemisorbed oxygen, and the other at high temperatures involving lattice oxygen [76].

## 5. Conclusions

Methane is the most abundant hydrocarbon in the atmosphere and is an important greenhouse gas, with a 21-fold greater relative radiative effectiveness than $CO_2$ on a per-molecule basis. Methane has been used in numerous studies since it is of great interest as a model compound for understanding the mechanisms of oxidation and catalytic combustion on metal oxides. All the described results in this paper represent clear evidence of the real potential of oxide and mixed oxide cerium-containing catalysts in achieving highly active catalysts in complete methane combustion as they start to emerge as valid alternatives to the more expensive and sensible noble metal catalysts. Significant advances have been achieved using metal oxides for total methane oxidation. These include the development of novel oxide and mixed oxide cerium-containing catalysts with appreciable catalytic reactivity. Shape structured oxides (nano-cubes, nanowires, nanobelts) have been widely investigated in recent years. Not only the nature of the exposed faces but also crystal defects (vacancies, interstitial atoms, shear planes) can have a decisive effect on the catalyst performances. In most cases, oxygen mobility is a key-factor in oxidation processes. Redox supports, especially those based on ceria, reinforce the performance of oxide catalysts in oxidation reactions by stabilizing the proper $M^{n+}/M^{(n+1)+}$ balance required for good activity.

The structural and textural characteristics of the ceria-based catalysts, which are determined by the method of preparation used, significantly influence their performance in methane combustion through their structural defects, particle morphology and surface area. On the other hand, the modification of ceria with one or several cations to obtain doped, supported or mixed oxide catalysts results in improved material reducibilities, due to the synergistic interactions between the different cationic species, leading to enhanced catalytic performance. Therefore, to obtain a highly active ceria-containing catalyst, showing a performance close to or even better than that of the noble metal-containing systems, further research should be performed in order to find the appropriate preparation method together with the right modifying cations and the optimum ratio between them.

These results create new challenges for the technology and for the total oxidation of methane. Looking ahead, the preparation of new catalysts will continue to advance as new methods and tools are available and as new paths appear on the horizon. These challenges and opportunities will require increasingly collaborative and interdisciplinary research to find solutions and to achieve full utilization.

**Author Contributions:** All authors contributed equally to this work. All authors have read and agreed to the published version of the manuscript.

**Funding:** This research received no external funding.

**Acknowledgments:** This work was supported by the University of Technology of Troyes, France. I. Fechete gratefully acknowledges the wonderful discussion of F. Garin, from Université de Strasbourg, concerning catalytic kinetics.

**Conflicts of Interest:** The authors declare no conflict of interest.

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
