# Peer review of "Total Oxidation of Methane on Oxide and Mixed Oxide Ceria-Containing Catalysts"

_catalysts, doi:10.3390/catal11040427_

Round 1

Reviewer 1 Report

The article will serve as a useful reference summary for researchers working in the field. Referencing is extensive, appropriate for a review article and adequate. The article can be published in "Catalysts" after minor revision. Below, I provide a list of minor suggested corrections, for the authors' consideration

line 106: "The complete oxidation of CH4 by its complete... "
               consider rephrasing the above sentence

line 115: "which is highly dispersed over support oxide catalysts.."

                Do you mean supported oxide catalysts?

line 124: "high poisoning ability"

    this is not an "ability"!!! You may like to consider one of the following
     1)....lack of ability to resisit in poisoning
     or
     2) ...high poisoning susceptibility 

line 168: "100000 h21"

     ???

Figure 2 caption: Is it meant Td?

lines 271-273: could be useful to state that such vacancies are self-charge-balanced and may occur w/o heterovalent cation exchange

line 384: ceria-supported?

lines 400-401: please rephrase

line 426: change "benefic" to "beneficial"

line 443: change "retains" to "retain"

line 515: change "obtained" to "obtain"

line 560: change "benefic" to "beneficial"

line 714: erase "science"

Line 893: "Moreover, in the more..."

       Consider: "Moreover, for the more..."

line 895: change "impact" to "impacts"

line 896: change "on hyfrothermal" to "on hydrothermal"

line 907: either "vapors are" or "vapor is"

line  925 : change "enhance the textural properties," to "improve the textural properties,"

line 961" consider changing "with" to "to an"

line 1038: activation energy

line 1137: change "require" to "requires"

line 1146: Consider starting with: "Doping the CuCeMgAlO with an additional

..."

line 1158: is it meant "metal free"?

lines 1164-1165: Please rephrase

line 1610: ref 192: delete "86."

Author Response

Dear Editor of “Catalysts” Journal,

We are pleased to submit the revised version of our manuscript, entitled “Total oxidation of methane on oxide and mixed oxide ceria-containing catalysts” by Marius Stoian, Vincent Rogé, Liliana Lazar, Thomas Maurer, Jacques C. Védrine, Ioan Cezar Marcu, Ioana Fechete. This revised manuscript, approved by all the authors, is not currently under evaluation by another journal.

We would like to express our sincere gratitude to the reviewers for reading the manuscript carefully, and we are grateful to them for their remarks on our work. We have modified the manuscript according to the comments of the reviewers with which we agree.

We thank the editors for their time and the attention paid to our manuscript.

Yours sincerely,

Ioana Fechete

Reviewer 2 Report

The review manuscript “Total oxidation of methane on oxide and mixed oxide cerium-supported catalysts” reports a comprehensive overview of the most recent findings concerning the development of ceria-based catalysts for the total oxidation of methane. In particular, the manuscript addresses the crucial role of oxygen vacancies due to the formation of solid solution structures on the reactivity of metal and metal oxide catalysts in comparison to the undoped CeO2 samples, through a systematic analysis of the reaction conditions, mechanisms and kinetics. The review encompasses the overview of a large number of catalytic materials and a thorough analysis of the most relevant literature findings in the field, being written in a sufficiently clear and readable form. Therefore, in my opinion, the manuscript may be considered for publication in Catalysts after an adequate revision aimed at improving the critical analysis of literature data. In particular, my suggestion is to arrange at the end of each paragraph a summary of the various catalyst formulations, with the relative experimental conditions and performances, perhaps in Table form, to help the reader in understanding in more details the relationships among catalyst formulation, catalytic performance, reaction mechanisms and kinetics. The authors are also encouraged to provide a final paragraph including the state of the art and a critical overview of the perspectives in the field.

Author Response

(The authors gave the same response as above.)

Round 2

Reviewer 2 Report

the revised version of the manuscript addresses in a satisfactory way the criticisms raised on the old version and then it can be considered for publication